# A panel of nanobodies recognizing conserved hidden clefts of all SARS-CoV-2 spike variants including Omicron

Ryota Maeda [1,2,16]✉, Junso Fujita [3,4,16], Yoshinobu Konishi [1], Yasuhiro Kazuma [1], Hiroyuki Yamazaki[2,5], Itsuki Anzai[6], Tokiko Watanabe[6], Keishi Yamaguchi[4], Kazuki Kasai [2], Kayoko Nagata[1], Yutaro Yamaoka[7], Kei Miyakawa[7], Akihide Ryo[7], Kotaro Shirakawa [1], Kei Sato [8,9,10], Fumiaki Makino [3,11], Yoshiharu Matsuura [12,13], Tsuyoshi Inoue[4], Akihiro Imura [2]✉, Keiichi Namba [3,14,15]✉ & Akifumi Takaori-Kondo [1]✉

We are amid the historic coronavirus infectious disease 2019 (COVID-19) pandemic. Imbalances in the accessibility of vaccines, medicines, and diagnostics among countries, regions, and populations, and those in war crises, have been problematic. Nanobodies are small, stable, customizable, and inexpensive to produce. Herein, we present a panel of nanobodies that can detect the spike proteins of five SARS-CoV-2 variants of concern (VOCs) including Omicron. Here we show via ELISA, lateral flow, kinetic, flow cytometric, microscopy, and Western blotting assays that our nanobodies can quantify the spike variants. This panel of nanobodies broadly neutralizes viral infection caused by pseudotyped and authentic SARS-CoV-2 VOCs. Structural analyses show that the P86 clone targets epitopes that are conserved yet unclassified on the receptor-binding domain (RBD) and contacts the N-terminal domain (NTD). Human antibodies rarely access both regions; consequently, the clone buries hidden crevasses of SARS-CoV-2 spike proteins that go undetected by conventional antibodies.

[1] Department of Haematology and Oncology, Graduate School of Medicine, Kyoto University, Kyoto 606-8507, Japan. [2] COGNANO Inc., Kyoto 601-1255, Japan. [3] Graduate School of Frontier Biosciences, Osaka University, Osaka 565-0871, Japan. [4] Graduate School of Pharmaceutical Sciences, Osaka University, Osaka 565-0871, Japan. [5] Shizuoka City Shizuoka Hospital, Shizuoka 420-8630, Japan. [6] Department of Molecular Virology, Research Institute for Microbial Diseases, Osaka University, Osaka 565-0871, Japan. [7] Department of Microbiology and Molecular Biodefense Research, Yokohama City University Graduate School of Medicine, Yokohama 236-0004, Japan. [8] Division of System Virology, Department of Infectious Disease Control, International Research Center for Infectious Diseases, The Institute of Medical Science, The University of Tokyo, Tokyo 108-8639, Japan. [9] Graduate School of Medicine, The University of Tokyo, Tokyo 113-0033, Japan. [10] CREST, Japan Science and Technology Agency, Saitama 332-0012, Japan. [11] JEOL Ltd., Tokyo 196-8558, Japan. [12] Centre for Infectious Disease Education and Research, Osaka University, Osaka 565-0871, Japan. [13] Laboratory of Virus Control, Research Institute for Microbial Diseases, Osaka University, Osaka 565-0871, Japan. [14] JEOL YOKOGUSHI Research Alliance Laboratories, Osaka University, Osaka 565-0871, Japan. [15] RIKEN Centre for Biosystems Dynamics Research and SPring-8 Centre, Osaka 565-0871, Japan. [16] These authors contributed equally: Ryota Maeda, Junso Fujita. ✉email: maeda@cognano.co.jp; akihiroimura@cognano.co.jp; keiichi@fbs.osaka-u.ac.jp; atakaori@kuhp.kyoto-u.ac.jp

We have been in the historic pandemic for more than 2 years (2019–2022): numbers of confirmed deaths and infections still increase on a weekly basis. Researchers have developed many antibodies as weapons to fight against COVID-19[1–7]; however, emerging SARS-CoV-2 variants of concern (VOCs) that carry mutations in the spike often escape the immune system and therapeutic antibodies[8–10]. These antibodies also serve as armour and shields, allowing us to survey infected individuals and monitor circumstances via antigen test kits, enzyme-linked immunosorbent assays (ELISAs), dot blotting and so forth. However, accessibility to diagnostic and monitoring kits is imbalanced worldwide because antibody-producing cells (e.g., hybridoma cells and engineered mammalian cells) are, in many cases, dominated by licenced companies that incur high costs of research, development, production, and distribution. Thus, we tried to create easily accessible and broadly neutralizing antibodies for SARS-CoV-2 VOCs.

Although many antibodies and nanobodies[11–13] against the spike, especially the receptor-binding domain (RBD), have been established, those that are broadly neutralizing and detect all current variants including Omicron have yet to be developed[14]. Few anti-SARS-CoV-2 spike *nanobodies* have been applied for immune assays, such as ELISAs and lateral flow and especially membrane dot blotting assays[15]. Here, we provide a panel of broadly neutralizing nanobodies that detect five SARS-CoV-2 VOCs. Our structural analyses show that two clones—P17 and P86—capped the receptor-binding domain (RBD) regardless of their up or down conformations. The most potent clone, P86, bridged a gap between the down conformation of the RBD and the N-terminal domain (NTD) of a neighbouring protomer. The epitopes on the RBD recognized by the two clones partially overlapped; in particular, the epitope recognized by P17 shifted away from the L452R mutation observed in the SARS-CoV-2 Delta variant spike. Moreover, the epitopes of P86 are conserved among the Omicron variants BA.1, BA.2 and BA.3. P86 potently neutralized the SARS-CoV-2 Omicron variants compared to clinically available therapeutic antibodies.

## Results

**Purifying a SARS-CoV-2 spike protein retaining the furin cleavage site.** The SARS-CoV-2 spike is one of the main targets for human antibodies (Fig. 1a, b)[16,17]. Its trimer complex is expressed on the surface of the virion and plays a role in fusion with host cells expressing the binding partner angiotensin-converting enzyme II (ACE2)[18,19]. The most featured alteration of the SARS-CoV-2 spike compared to other coronavirus spikes is the generation of a furin cleavage site (-R-R-A-R[685]-: furin protease hydrolyses the C-terminal peptide bond of R[685])[20] located directly between the S1 (residues 1–685) and S2 (residues 686–1213) domains[17,21]. Because it was difficult to obtain the extracellular domain of SARS-CoV-2 spike with the furin site from the supernatant of human embryonic kidney (HEK) 293T cells, we attempted to purify it from cell lysate. We obtained a large complex of SARS-CoV-2 spike with detergents used in crystallizing membrane proteins[22] (see Supplementary Fig. 1a–c and 'Methods'). As reported previously, although peaks were broad, buffers containing high salt and detergent concentrations maintained the SARS-CoV-2 spike trimer assembly[23] (see Supplementary Fig. 1b). We noticed that a high salt concentration was necessary to stabilize the SARS-CoV-2 spike complexes[24] (Supplementary Fig. 1b).

**Immunization of alpacas and selection of nanobodies.** To obtain spike-specific nanobodies, we immunized two alpacas with the purified whole extracellular domains of the SARS-CoV-2

spike complex from cell lysates[25] (see Supplementary Fig. 1c and 'Methods'). Serum titres of alpacas for SARS-CoV-2 spike increased and reached a plateau, indicating that both alpacas were effectively vaccinated (see Supplementary Fig. 1d). We performed one round of biopanning with different proteins: the whole extracellular domain, the S2 domain alone of the SARS-CoV-2 spike, or the whole extracellular domain of the seasonal cold coronavirus HCoV-OC43 spike[26,27] (see 'Methods'). After deep-sequencing nanobody-coding genes of each panned sublibrary, we identified 18 clones that were significantly enriched in sublibraries panned with the whole extracellular domain of SARS-CoV-2 compared with those panned with the others[28] (Fig. 1c, d and see 'Methods'). We were able to express nine clones in bacteria and purified them as C-terminally His-tagged dimers (see Supplementary Fig. 2); we then characterized their avidity, applicability, and neutralizing activity, as described below.

**Kinetics and detectability of nanobody binding.** First, we measured the binding kinetics of these clones for the whole extracellular domain of SARS-CoV-2 spike, which was the same construct that was used for immunization, via biolayer interferometry (Fig. 1e). We noticed that the salt concentrations in the kinetic assay buffers affected the avidities of the clones. Of note, high salt concentrations were necessary to stabilize the spike complex (see Supplementary Fig. 1b). Four clones (P17, P86, C116 and P334) showed high avidity for the SARS-CoV-2 spike complex in hypertonic buffer; the others (C17, C49, P158, C246 and P543) showed high avidity in hypotonic buffer (see Supplementary Fig. 3).

Second, we checked these nanobodies via microscopy observation using HEK293T cells expressing the SARS-CoV-2 spike. Seven clones stained near the cell surface; two clones (C49 and C246) stained only an intracellular region (see Supplementary Fig. 4a, b). Third, flow cytometric analysis supported this observation: the seven clones detected the spike protein on the cell surface, but the C49 and C246 clones did not (see Supplementary Fig. 4c). These results suggested that each clone recognizes the SARS-CoV-2 spike protein in a status-dependent manner.

**Sensitivities of the clones for the RBD mutations of VOCs.** The World Health Organization (WHO) has declared five strains to be VOCs, which have mutations in the RBD: Alpha/UK/B.1.1.7 (N501Y); Beta/South Africa/B.1.351 (K417N, E484K and N501Y); Gamma/Brazil/P.1 (K417T, E484K and N501Y); Delta/India/B.1.617.2 (L452R and T478K)[29,30]; and—after submitting the original manuscript—Omicron/Southern Africa/B.1.1.592[31]. The SARS-CoV-2 virus strains having mutations in the residues E484 or L452 frequently escape neutralizing antibodies[32–34]. We checked which clones could detect which spike variants via microscopy observation. Two clones (C17 and P17) exhibited severely reduced sensitivity for the spike variant carrying the E484K mutation; previously reported SARS72[35], mNb6[36] and Ty1[37] nanobodies also did not detect these variants. Although the Ty1 nanobody did not sense the spike variant having the L452R mutation, the clones provided in this study remained able to detect the L452R substitution (see Supplementary Fig. 5).

**Nanobodies for test kits.** The finding that some clones recognized fluctuating, immature, or possibly unfolded SARS-CoV-2 spike proteins led us to consider whether the clones could detect the denatured SARS-CoV-2 spike variants via immunological assays. We sampled cell lysates expressing full-length C-terminally C9-tagged SARS-CoV-2 spike variants in Laemmli's sodium dodecyl sulfate (SDS) buffer under reducing conditions[38]: The

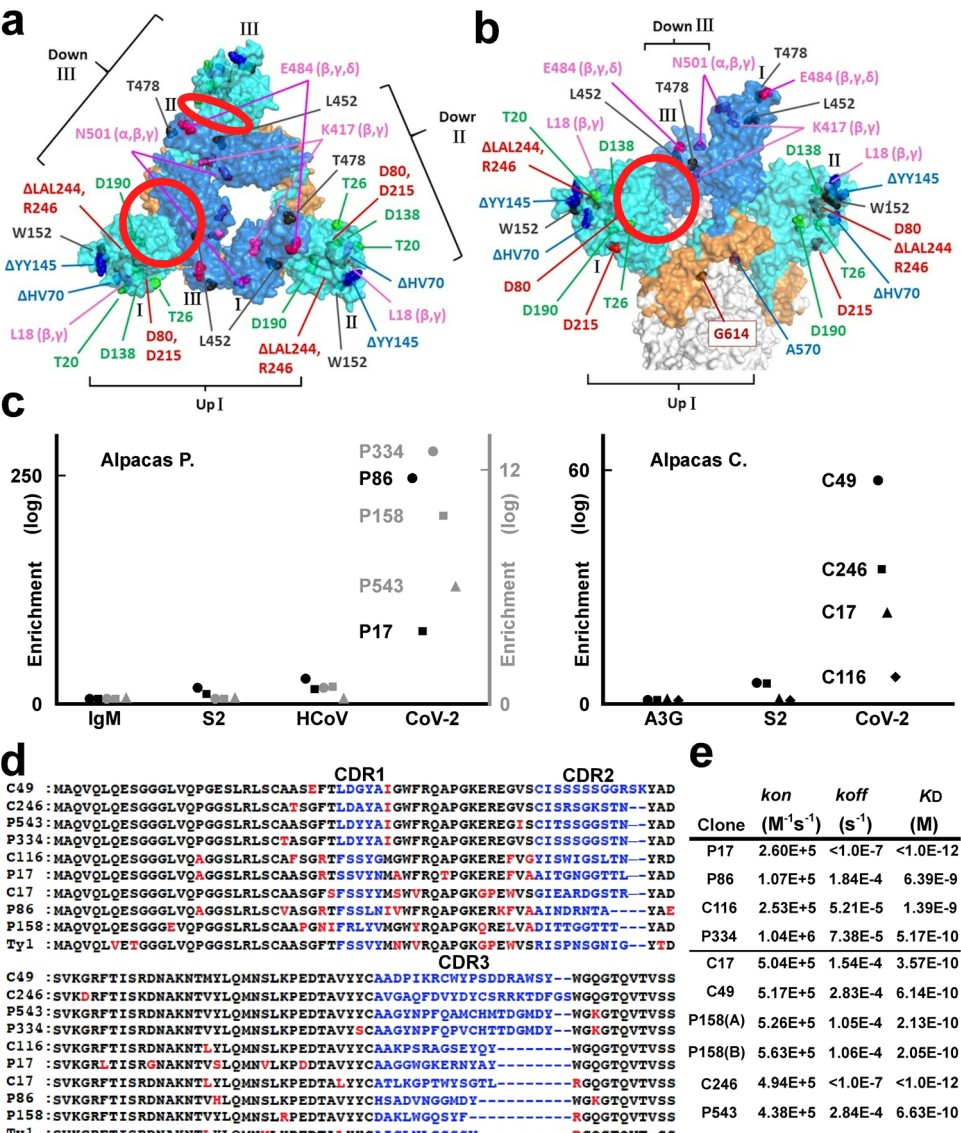

**Fig. 1 Structural features of the SARS-CoV-2 spike and the panel of nanobodies. a, b** Top (**a**) and side (**b**) views of the 1-up+2-down conformation of the SARS-CoV-2 spike trimer[95] (PDB entry: 7KRR); domains of each protomer numbered I (up), II (down) and III (down). The NTD, the RBD and the C-terminal domain (CTD) of the SARS-CoV-2 S1 domain are coloured cyan, blue, and orange, respectively; mutations and deletions studied here are indicated; clefts between the down-RBD and the NTD of the neighbouring protomer are highlighted with red circles. **c** Graphs showing existence ratios (two different scales drawn black and grey)—$P$ values in $\log_{10}$ are plotted (see 'Methods')—after panning with IgM; APOBEC3G (A3G)[26]; the S2 domain (S2); the HCoV-OC43 (HCoV) spike[27]; and the SARS-CoV-2 spike (CoV-2). **d** Sequences of nine clones and that of Ty1 are compared[37]: changed amino acids in the framework regions are red; the variable complementary-determining regions (CDRs) are shown in blue. **e** Calculated binding kinetics between each nanobody and the SARS-CoV-2 spike from sensorgrams (see Supplementary Fig. 3) are summarized. The upper four clones were assayed in a high-salt buffer ("stabilized" spike); the others were assayed in an anionic buffer ("fluctuated" spike).

P158, P334 and P543 clones specifically detected SARS-CoV-2 spike proteins with the same sensitivity as the mouse monoclonal C9-tagged antibody (Fig. 2a). The banding patterns suggested that each clone recognized a part of the S1 domain. These clones (P158, P334 and P543) detected SARS-CoV-2 spike variants—including Alpha and Beta variants—via western blotting.

**Nanobody-based ELISA and lateral flow assay.** We then tried to develop a sandwich ELISA kit to detect the SARS-CoV-2 spike using these nanobodies. We tested eight clones as detector nanobodies with six blocking conditions on P158-coated plates; we found that P86 and P543 worked well (see Supplementary Fig. 6). An ELISA kit, using P158 as the capture nanobody and P543 as the detection nanobody detected the SARS-CoV-2 spike

(Fig. 2b). Both P543 and P86 were able to specifically detect the SARS-CoV-2 spike original and Beta variant in cellular homogenates: HEK293T cells expressing the full-length spikes were lysed and sequentially diluted (Fig. 2c). When the detection antibody was changed to Fc-tagged P543 (P543-Fc), the detection limit reached below 20 ng ml⁻¹ (Fig. 2d), and the sensitivity was the same as that of an ELISA kit used for the detection of hae-magglutinin of influenza virus[39,40]. Moreover, antigen test kits based on a lateral flow membrane assay in which P158 lined a nitrocellulose membrane and P86 was conjugated to labelled beads were also able to detect 300 ng of the spike Delta variant (Fig. 2e). We checked the capability of each clone via microscopy observations using cells expressing SARS-CoV-2 spike variants carrying associated mutations. The clones that were useful for

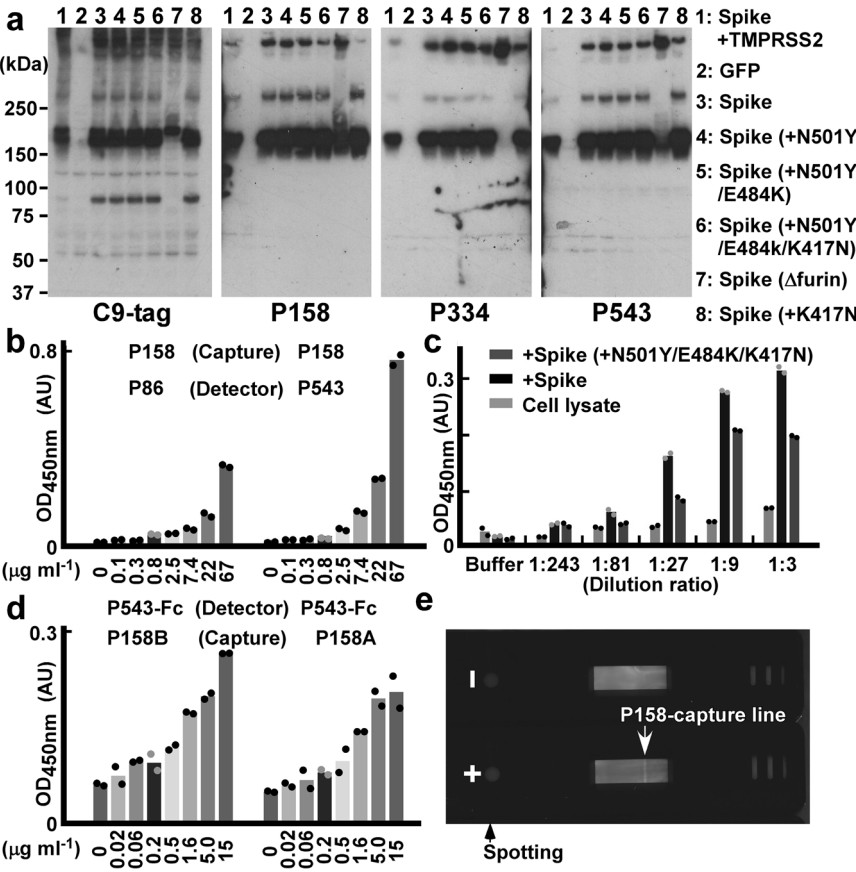

**Fig. 2 Nanobodies for western blotting assays, ELISAs and lateral flow assays. a** Nanobody-based western blotting assays of SARS-CoV-2 spikes: lysates of HEK293T cells expressing GFP or C-terminally C9-tagged SARS-CoV-2 spike (D614G) variants (indicated with numbers) were blotted with the indicated nanobodies. **b**–**d** Sandwich ELISA: **b** signals of gradually diluted sample concentrations using P158 as the capture antibody and P86 or P543 (**b**, **c**) or P543-Fc (**d**) as the detection antibodies are graphed as a mean of the measurements from two technical replicates. In **c**, sequentially diluted lysates of HEK293T cells expressing the original SARS-CoV-2 spike (D614G) or Beta variant (D614G, N501Y, E484K and K417N) were assayed. Data are representative of one independent experiment out of two biologically independent experiments. **e** A photograph of the results from lateral flow nitrocellulose membrane assays: P158 for the capture line (allow head) and P86 for the detection beads; 300 ng of the purified SARS-CoV-2 spike Delta variant (D614G, L452R and T478K) (+) or PBS (−) was spotted.

blotting (P158, P334 and P543) were still able to detect the SARS-CoV-2 spike carrying mutations in the S1 domain, including deletion mutations (del) of H69, V70, Y144, Y145, L242, A243 and L244 and point mutations of L18F, T20N, P26S, D138Y, R190S, D215G, R246I, N439Y, Y453F, T478K and A570D[41–44] (see Supplementary Fig. 7).

**Detecting the SARS-CoV-2 Omicron spike variant**. After submission of the original manuscript (10 November 2021), a new SARS-CoV-2 variant (Omicron) was reported (22 November 2021) and declared the fifth VOC (26 November 2021)[31,45]. The Omicron variant (B.1.1.529 and BA lineages) had become the most recognized VOC and outcompeted the Delta variant[46]. Under these circumstances, we assessed whether our nanobodies could detect the SARS-CoV-2 Omicron variant spike protein. First, we tested which nanobodies could detect the Omicron spike protein via western blotting and found that P86, C246, P158, P543 and P334 could detect both the original and Omicron variant spike proteins (Fig. 3a). Second, using flow cytometry assays, we reconfirmed that P17, P86, P158, P334 and P543 could recognize the Omicron spike on the cell surface (Fig. 3b). Therefore, we tested the ability of the kits made with these nanobodies to detect SARS-CoV-2 spike proteins from nasal swab specimens.

**Nanobody-based test kits for real nasal swab specimens**. We sampled 22 swab specimens from emergency fever patients. Sixteen samples were confirmed to be SARS-CoV-2 positive by PCR. We compared these samples to samples from healthy volunteers with P158-based western blotting and observed smeared and accumulated bands in swab specimens from the patients (Fig. 4a). The P158 nanobody detected bands in the 16 patient samples (Fig. 4b) with molecular sizes similar to those of transfected spike proteins (Fig. 3a).

Next, we developed nanobody-based lateral flow assay kits for nasal swab specimens. We used P158-dimer for the capture line and conjugated the P543-trimer to fluorescent beads. We sampled swab specimens by adding detergents and nuclease (see 'Methods'). We observed positive lines in the nasal swab specimens from the patients but not in those from the healthy volunteers (Fig. 4c). For the detection of spike proteins in nasal swab specimens using sandwich ELISA, we used a Flag-tagged P86-trimer as the detection antibody (the capture nanobody P158-dimer was His-tagged). The detection limit was below 16 ng ml⁻¹. Thus, this nanobody-based ELISA kit was able to detect spike proteins in the nasal swab specimens from patients (Fig. 4d).

**Neutralizing activity of the nanobodies against SARS-CoV-2 spike variants**. We assayed whether these clones could neutralize

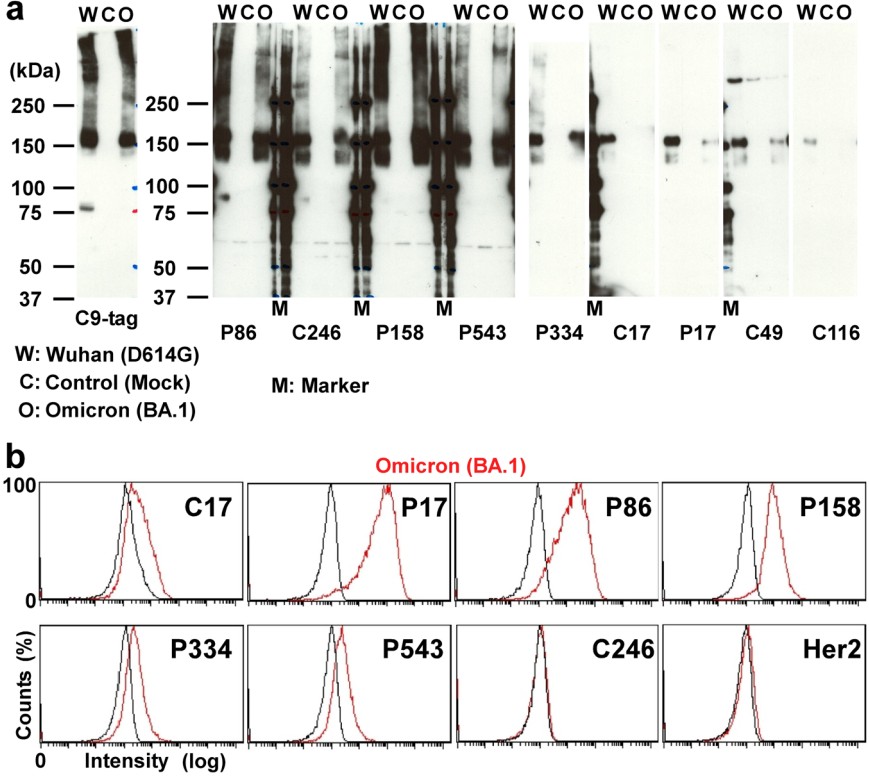

**Fig. 3 Nanobodies against the SARS-CoV-2 Omicron variant. a** Nanobody-based western blotting assays of SARS-CoV-2 spikes. HEK293T cell lysates (C: as control of mock transfection) expressing the C-terminally C9-tagged SARS-CoV-2 spike proteins (W: Wuhan-1) and one of the Omicron variant (O: Omicron BA.1) were blotted with an anti-C9 antibody and the indicated nanobodies. **b** Flow cytometric analyses of DT40 cells. Log scaled signals (x-axis) of the indicated eight nanobodies with parent cells (black) and cells stably expressing the SARS-CoV-2 Omicron spike protein (red) were counted (y-axis) and plotted.

virus infection using SARS-CoV-2 spike pseudotyped HIV-1-based viruses. We used K562 cells stably expressing human ACE2 and serine protease TMPRSS2. We also produced pseudotyped viruses encoding the luciferase gene and expressing the SARS-CoV-2 spike variants. Five (C17, P17, P86, C116 and C246) out of nine clones inhibited the pseudotyped virus expressing the spike (original WH-1: D614G) in a dose-dependent manner (Fig. 5a). Then, we also tested the potency of the five clones for the other SARS-CoV-2 variants—Alpha, Beta and Delta. The P17, P86 and C116 clones more potently neutralized the Alpha variant than Ty1; the P86 and C246 clones significantly suppressed the Beta variant; and P17 and C246 suppressed the Delta variant (Fig. 5a). The $IC_{50}$ results were summarized and compared: C246, which stained intracellular spike (see Supplementary Fig. 5c), suppressed four variants tested; P86 neutralized the Beta but not Delta variants; and P17 neutralized the Delta but not Beta variants (Fig. 5b). Because it was unclear to us why P17 could be affected by the E484K mutation (Beta) and P86 by the L452R mutation (Delta), we next determined the differences in the epitopes of the two clones.

**The binding epitopes of the P86 and P17 clones**. To determine the epitopes of the P86 and P17 clones on the SARS-CoV-2 spike, we performed a single-particle analysis of the spike-P86 and spike-P17 complexes using cryo-electron microscopy (cryo-EM). We observed 2-RBD-up (2-up+P86; 46.2%) and 3-RBD-up (3-up+P86; 53.8%) from the spike-P86 complex and 1-RBD-up (1-up+P17) and 2-RBD-up (2-up+P17) from the spike-P17 complex (Fig. 6a, b, Table 1, and see Supplementary Fig. 8). The densities of P86 and P17 were observed just on the outside surfaces of both up- and down-RBDs—this arrangement was more apparent for

the down-RBD (see Supplementary Fig. 8). We observed that the densities of the clones bound to the down-RBD extended to the NTD of the neighbouring protomer (PDB entry: 7KRR)—more apparently in P86—the densities of P17 were too weak to determine the orientation (Fig. 6c, d).

To build an atomic model of 2-up+P86, we determined the crystal structure of P86 at 1.60 Å resolution (Table 2). The asymmetric unit contained two structurally identical monomers (a root mean square deviation for $C_{\alpha}$ atoms of 0.20 Å); the three CDR regions were visualized well (see Supplementary Fig. 9a, b). We fit the P86 crystal structure to the density near the down-RBD in a locally refined map. The map showed two featured regions: a loop-like density between the down-RBD and the NTD of the neighbouring protomer and an elongated density located on the top (see Supplementary Fig. 9c). We fit the CDR3 and the C-terminal (C-ter.) region into the former and the latter, respectively, and the whole body was adequately accommodated. After manual model modifications, the P86 monomer fit well into the final sharpened map (Fig. 6c and see Supplementary Fig. 9c–e). The edge of the CDR3 region was inserted into the cleft between the down-RBD and the NTD of the neighbouring protomer, forming many hydrophilic and hydrophobic interactions (Fig. 6c [right panel]). Compared to the density of P86, the density of P17 shifted towards the top of the RBD (Fig. 6d). Therefore, the interface between the down-RBD and P17 was near the receptor-binding motif (RBM: ACE2-binding region); on the other hand, the interface between the down-RBD and P86 shifted to the NTD of the neighbouring protomer (Fig. 6e).

To date, human antibodies target four epitopes, all on the RBD: Class-1 (RBM class 1), Class-2 (RBM class 2), Class-3 (RBD core 1) and Class-4 (RBD core 2)[47–49] (see Supplementary Fig. 9f). The

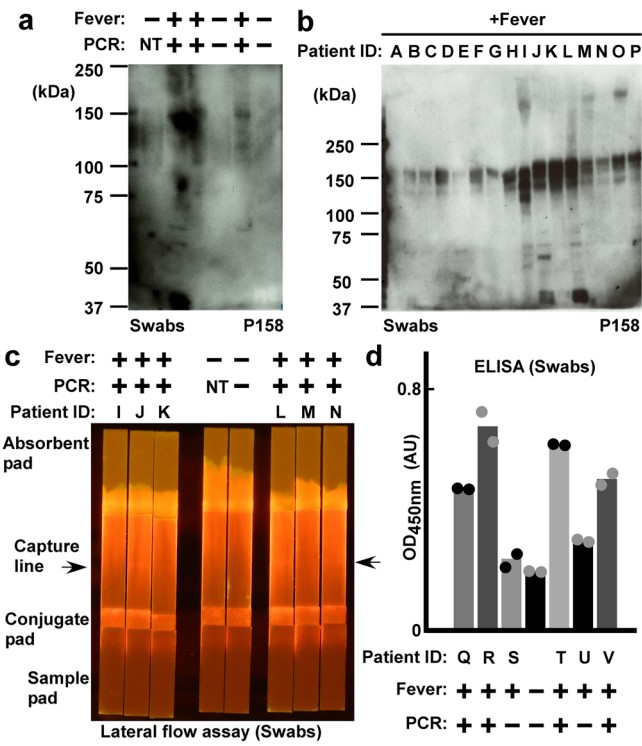

**Fig. 4 Nanobodies for immune assays testing nasal swab specimens.**
**a** P158 nanobody-based western blotting of nasal swab specimens. Patients with fever (+) and confirmed to be positive for SARS-CoV-2 by PCR test (+); healthy volunteers with no fever (−) and negative by PCR test (−) were compared (NT: not tested). **b** P158 nanobody-based western blotting assays of swabs from emergency fever patients. A to D and I to P were confirmed positive for SARS-CoV-2 by PCR test; E to H were not confirmed to be positive by the PCR test. **c** P158 nanobody-lined lateral flow assays of swabs from emergency fever patients. Control and patient samples are the same as those in **a** and **b**, respectively. **d** P158 nanobody-captured ELISA. Means of detection signals of the nasal swab specimens are shown. Data are representative of one independent experiment out of two biologically independent experiments.

epitope of P86 was located outside of Class-2 and Class-3 and was on the opposite side of Class-1 and Class-4 (Fig. 6f and see Supplementary Fig. 9f). We compared two antibodies—DH1041(PDB entry: 7LAA) and Ty1 (PDB entry: 6ZXN)—that target the near-surface of the down-RBD where P86 is bound[37,50]. We superposed them on the single down-RBD within the 2-up+P86 model, which clearly showed P86—especially the CDR1 and CDR3 loops—uniquely bound the down-RBD and the NTD of the neighbouring protomer (Fig. 6c, g). We noticed that the P86-bound down-RBD interfered with the ACE2 in its access to the neighbouring up-RBD (see Supplementary Fig. 9d). It seems that P86 is small enough to access the *hidden* cleft that is not recognized by human antibodies[51] (see Supplementary Fig. 10).

**Epitope mapping of P86 to the Omicron variant.** The Omicron variants have spread rapidly. To date, several Omicron lineages have been classified: BA.1, BA.2 and BA.3[46,52] (Fig. 7). Recent works warn that almost all neutralizing antibodies on the market or under development do not have the activity against the Omicron variants, except for sotrovimab (S309), which maintains the neutralizing activity against BA.1[53–57]. In addition, several research groups have reported that the neutralizing activity of sotrovimab is significantly reduced against BA.2[58,59].

We mapped the mutations of the Omicron variants on the 2-up +P86 structure (Fig. 7a,b) and evaluated how each mutation would affect the nanobody binding[60,61] (see Supplementary Table 1). While the L452R mutation in the Delta variant was the most effective in reducing the nanobody binding ($\Delta G = +1.0$ kcal mol$^{-1}$), any mutations in BA.1 and BA.2 (including BA.3) would not affect the binding of P86. In particular, the variants with Q493R, G496S, and Q498R mutations bind to ACE2[62,63] were placed around P86 without contributing to the binding to P86 (Fig. 7c and see Supplementary Table 1). Finally, P86 also contacted the inner side of the NTD, where no mutations have been seen in any VOCs (Fig. 7a, b and see Supplementary Table 1).

**Neutralizing activity of the nanobodies against the SARS-CoV-2 Omicron variants.** We assessed whether P86 neutralizes the SARS-CoV-2 Omicron variants using pseudotyped and authentic viruses. As a control, we included sotrovimab (S309) in these assays. P86 neutralized both the BA.1 and BA.2 Omicron pseudotyped viruses, while sotrovimab showed a reduction in its ability to suppress the BA.2 Omicron variant (Fig. 8a).

Finally, we evaluated the neutralizing activities of P86 and C246 using the authentic Omicron BA.1 variant virus (strain hCoV-19/Japan/TY38-873/2021) (Fig. 8b–d). Although C246 could not neutralize the virus even at the highest concentration tested (25 μg ml$^{-1}$) (Fig. 8d), P86 suppressed plaque formation of the authentic Omicron BA.1 variant virus (IC$_{50}$ of 0.18 μg ml$^{-1}$) with an efficiency similar to that of sotrovimab (IC$_{50}$ of 0.52 μg ml$^{-1}$) (Fig. 8b, c).

## Discussion

We provide the sequences of the nanobodies recognizing recently emerging SARS-CoV-2 spike VOCs including the Omicron variants. Only the sequence information or expression vectors are sufficient to produce these clones in basic biological laboratories. We showed that the nanobodies detected the spike variants via ELISA, immunochromatography, and blotting assays. Therefore, these clones can be applied for surveillance of the virus in wastewater, monitoring infected individuals (e.g., passengers, farm animals, or pets), and self-testing. A limitation of this study is that the C246 clone could not neutralize the authentic SARS-CoV-2 virus infection. These results signify that antibodies and chemicals neutralizing pseudotyped viruses cannot necessarily neutralize real viruses[64–66].

The epitopes of P17 and P86 were shifted relative to each other (Fig. 6e), which caused different tolerances to the mutations—P17 neutralized the Delta (L452R) variant, whereas P86 neutralized the Beta (E484K) variant. The L452R variant may substantially reduce the avidity of P86 to the RBD: R452 is located near the centre of the P86 binding interface. By contrast, because P17 binds to the surface near the top of the RBD and R452 exists on the edge of the interface, P17 tolerates the long R452 side chain. E484 is part of the flexible *hook* rope of the RBD. Structural and molecular dynamics studies have suggested that the E484K mutation disorders the hook region[67]. We assume that the Class-2 antibodies can only bind a solid state of the hook containing E484, whereas P86 is not severely influenced by states of the hook region containing K484 because P86 also contacts the conserved cleft between the RBD and the NTD of the neighbouring protomer. The epitope of P86 seems too narrow to be accessed by conventional antibodies. Indeed, the region of the RBD where P86 binds has not yet been classified as an epitope of human antibodies[68] (Fig. 6f and see Supplementary Figs. 9 and 10). Therefore, we expect these clones to be *lime in a gimlet* of antibody-based therapies.

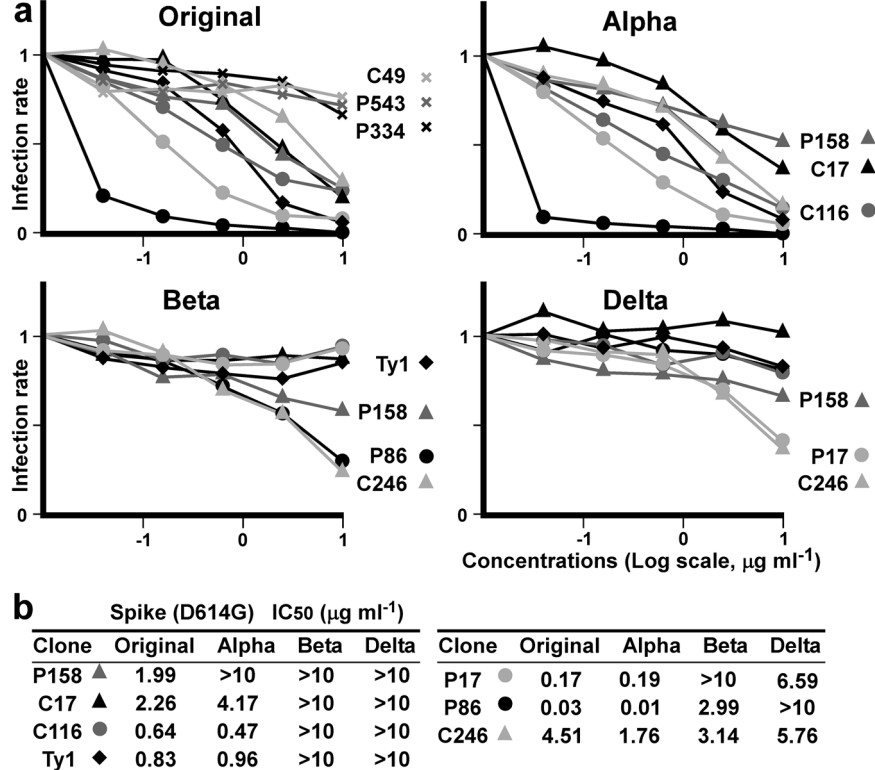

**Fig. 5 Pseudotyped virus neutralization assay. a** Means of infecting units of virus for the indicated SARS-CoV-2 spike variants are graphed. K562 cells expressing ACE2 and TMPRSS2 infected with HIV-1-based pseudotyped virus were assessed via luciferase assay. Relative infection units at the indicated concentrations of the nanobodies, which are marked individually, are plotted. Data points are mean fold changes of technical replicates and are representative of at least two independent experiments. **b** Measured IC50s for each variant are summarized.

After submitting the original manuscript, the Omicron variants have emerged, spread, and outcompeted the Delta variant. The Omicron variants escaped from the sera of vaccinated or previously infected people and evaded almost all therapeutic antibodies. Here, we presented a panel of nanobodies that can detect the Omicron spike variants in nasal swab specimens and neutralize the SARS-CoV-2 Omicron variants as potentially universal therapeutics of COVID-19.

## Methods

**Ethics statement for animal care and informed consent**. Two young alpacas (*Vicugna pacos*) half-siblings—a 19-month-old male named "Puta" and a 19-month-old female named "Christy"—were immunized. Veterinarians of the KYODOKEN Institute for Animal Science Research and Development (Kyoto, Japan) bred, maintained health, recorded conditions, and performed the immunization studies by adhering to the published Guidelines for Proper Conduct of Animal Experiments by the Science Council of Japan. The KYODOKEN Institutional Animal Care and Use Committee approved the protocols for these studies (approval number 20200312) and monitored health conditions. The veterinarians immunized the alpacas with antigens and collected blood samples under anaesthesia. Nasal swab specimens were collected from patients who had agreed to use them for these studies. The Committee of Shizuoka City Hospital approved the protocols (approval number 20220128).

**Immunization and library generation**. Immunized antigens were purified recombinant SARS-CoV-2 spike complexes: the extracellular domain of the original Wuhan-1 protein (GenBank: QHD43416) with or without the D614G mutation that carried a maintained or mutated furin cleavage site—N[679]SPRRA or IL[680]. The protein mixture emulsified in complete Freund's adjuvant was subcutaneously injected into the two alpacas up to 9 times at 2-week intervals. Blood samples were collected from the jugular vein; peripheral blood mononuclear cells (PBMCs) were obtained with a sucrose density gradient using Ficoll[69] (Nacalai Tesque, Kyoto, Japan). The PBMC samples were washed with PBS and suspended in RNAlater solution (Thermo Fisher Scientific K.K., Tokyo, Japan). Total RNA was isolated from the PBMC samples (Direct-Zol RNA MiniPrep: Zymo Research, Irvine, CA).

Complementary DNA was synthesized from 1 μg of the total RNA as a template with random hexamer primers and using SuperScript II reverse transcriptase (Thermo). Coding regions of the heavy-chain variable domains were amplified using LA Taq polymerase (TAKARA Bio Inc., Shiga, Japan) with two PAGE-purified primers (CALL001: 5′-GTCCTGGCTGCTCTTCTACAAGG-3′ and CALL002: 5′-GGTACGTGCTGTTGAACTGTTCC-3′). The amplified coding gene fragments of heavy-chain variable domains were separated on a 1.5% low-temperature melting agarose gel (Lonza Group AG, Basel, Switzerland). Approximately 700 base pair bands corresponding to the heavy-chain only immunoglobulin were extracted (QIAquick Gel Extraction Kit: Qiagen K.K., Tokyo, Japan). Nested PCR was performed to amplify coding genes of VHH domains using the VHH-PstI-For and VHH-BstEII-Rev primers and subcloned into the pMES4 phagemid vector[12,70] (GeneArt DNA Synthesis: Thermo). Electroporation-competent *Escherichia coli* TG1 cells (Agilent Technologies Japan, Ltd., Tokyo, Japan) were transformed with the ligated plasmids under chilled conditions (Bio-Rad Laboratories, Inc., Hercules, CA). Colony-forming units of the libraries were checked with limiting dilution to maintain >10[7] per microlitre. Colonies from 8 ml of cultured cells were harvested, pooled, and reserved in frozen glycerol stocks as parent libraries.

**Plasmid construction for protein expression**. The gene encoding the extracellular domain of the SARS-CoV-2 spike protein (residues 31–1213) was codon-optimized and synthesized into the pcDNA3.1(+) vector (Thermo) with an N-terminally modified IL-2-derived signal peptide[71] (ILco2: MRRMQLLLLIAL-SLALVTNS); proline substitutions at residues K986P and V987P; and the C-terminal T4-phage fibritin trimerization domain (foldon) following a 6×His-tag[34,72–74]. The expression vector of the SARS-CoV-2 S2 domain (residues 744–1213) was constructed by removing an N-terminal part of the extracellular domain of the SARS-CoV-2 spike (residues 31–743) and subcloned into the pcDNA3.1(+) vector. The expression vectors of the full-length SARS-CoV-2 spike including variants and the human serine protease TMPRSS2 with a C-terminal C9-tag (TETSQVAPA) were acquired from AddGene (Summit Pharmaceutical International, Tokyo, Japan). The genes encoding the extracellular domain of human ACE2 (residues 1–614) and the endemic human coronavirus HCoV-OC43 (residues 1–1322) were codon-optimized, synthesized with a C-terminal 6×His-tag and subcloned into the pcDNA3.1(+) vector. The whole gene of the human apolipoprotein B messenger-RNA-editing enzyme catalytic polypeptide-like (APOBEC) 3G (A3G)[26] was also codon-optimized, synthesized with a C-terminal 6×His-tag, and subcloned into the pcDNA3.1(+) vector.

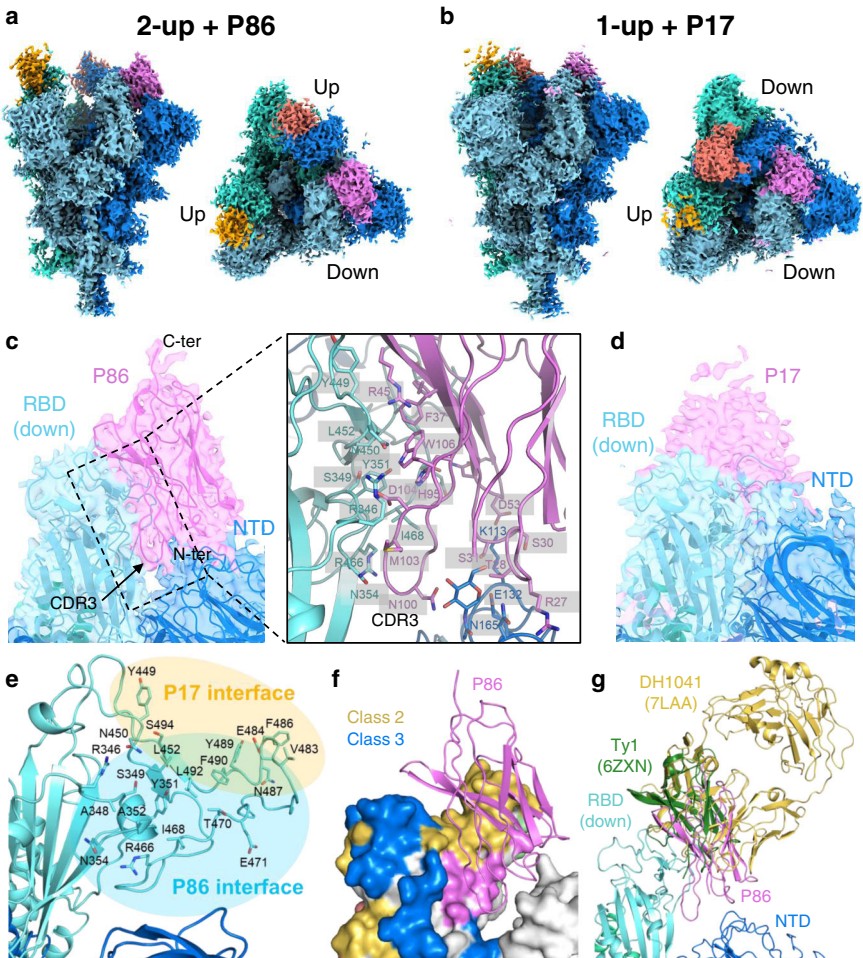

**Fig. 6 Cryo-EM density maps of SARS-CoV-2 spike-P86 and SARS-CoV-2 spike-P17 complexes. a, b** Final sharpened maps of the (**a**) 2-up+P86 and (**b**) 1-up+P17 datasets. The map regions corresponding to one protomer in the spike protein are coloured differently in cyan, blue, and turquoise. The map regions corresponding to one P86 or P17 molecule bound to the RBD regions are coloured magenta, red, and orange, respectively. **c** Close-up view around P86 bound to the down-RBD with our fitted model. Right panel shows close-up view around the interface among P86, the down-RBD, and the NTD of the neighbouring protomer. **d** Close-up view around P17 bound to the down-RBD. The model of D614G spike trimer (PDB entry: 7KRR) was fitted on the maps[95]. **e** The residues located on the interface between the down-RBD and P86 or P17. **f** Epitope mapping on the down-RBD. The epitopes of Class-2, Class-3 and P86 are shown in yellow, blue, and magenta, respectively. The colouring priority of the overlapping region is Class-3 > Class-2 > P86. The model of P86 is a ribbon structure. **g** Structure comparison with other antibodies bound to the down-RBD. The structures are superposed on the single RBD region.

For structural analyses, the sequence encoding the spike ectodomain (residues 1–1208) with proline substitutions, a "GSAS" substitution at the furin cleavage site (residues 682–685)[75], and the C-terminal foldon trimerization motif followed by an 8×His-tag was cloned into the pcDNA3.1(+) expression vector. Furthermore, the D614G mutation in the spike protein was introduced by the inverse PCR method. For cryo-EM, recombinant spike proteins were transiently expressed in Expi293-F cells (Thermo) maintained in HE400AZ medium (Gmep, Inc., Fukuoka, Japan). The expression vector was transfected using a Gxpress 293 Transfection Kit (Gmep) according to the manufacturer's protocol. The culture supernatants were harvested 5 days post-transfection. The C-terminally 6×His-tagged spike proteins were purified using a nickel Sepharose 6 FF column (Cytiva) and size exclusion chromatography using a Superdex200increase 10/300 GL column (Cytiva) with buffer containing 50 mM HEPES (pH 7.0) and 200 mM NaCl.

**Biopanning**. The purified proteins were (i) the extracellular domain of the SARS-CoV-2 spike; (ii) only the S2 domain of the SARS-CoV-2 spike; (iii) the extracellular domain of the seasonal cold coronavirus spike of HCoV-OC43; (iv) A3G; and (v) homemade IgM coupled to N-hydroxysuccinimide (NHS)-activated magnet beads (Dynabeads, Thermo). One round of biopanning was performed using different protein-coated magnet beads in 50 mM phosphate buffer (pH 7.4) containing 1% n-dodecyl-β-D-maltopyranoside (DDM: Nacalai), 0.1% 3-[(3-cholamidopropyl)dimethylammonio]-1-propane sulfonate (CHAPS: Nacalai), 0.001% cholesterol hydrogen succinate (CHS: Tokyo Chemical Industry Co., Ltd. (TCI), Tokyo, Japan), 0.1% LMNG (Anatrace, Maumee, OH) and 500 mM NaCl. After 3 washes with the same buffer, the remaining phages bound to the washed beads

were eluted with a trypsin-ethylenediaminetetraacetic acid (EDTA: Nacalai) solution at room temperature for 30 min. The elution was neutralized with a PBS-diluted protein inhibitor cocktail (cOmplete, EDTA-free, protease inhibitor cocktail tablets: Roche Diagnostics GmbH, Mannheim, Germany) and used to infect electroporation-competent cells. The infected cells were cultured and selected in LB broth Miller containing 100 μg ml⁻¹ ampicillin (Nacalai) at 37 °C overnight. The selected phagemids were collected using a QIAprep Miniprep Kit (Qiagen).

**Sequence analysis of nanobody libraries**. The VHH-coding regions within parent libraries and 1-round target-enriched sublibraries were PCR amplified and purified using AMPure XP beads (Beckman Coulter, High Wycombe, UK). Then, dual-indexed libraries were prepared and sequenced on an Illumina MiSeq (Illumina, San Diego, CA) using a MiSeq Reagent Kit v3 with paired-end 300 bp reads[76] (Bioengineering Lab. Co., Ltd., Kanagawa, Japan). Approximately 100,000 paired reads of each library were generated. The raw data of reads were trimmed of the adaptor sequence using cutadapt v1.18[77], and low-quality reads were subsequently removed using Trimmomatic v0.39[78]. The remaining paired reads were merged using fastq-join[79] and then translated to the amino acid sequences using EMBOSS v6.6.0.0[80]. Finally, unique amino acid sequences in each library were counted using a custom Python script combining seqkit v0.10.1[81] and usearch v.11[82]. Enrichment scores of each clone were analysed by calculating the $P$-value of $\chi^2$ tests between the existing ratios among the different sublibraries.

We clustered nanobody sequences with no more than six Damerau–Levenshtein distances[83,84]. We found eight and ten clusters—from Cristy's and Puta's libraries, respectively—significantly enriched in only the SARS-

**Table 1 Cryo-EM data collection and processing.**

| Dataset | 2-up+P86 (EMDB-32078) (PDB ID: 7VQ0) | 3-up+P86 (EMDB-32079) | 1-up+P17 (EMDB-32080) | 2-up+P17 (EMDB-32081) |
|---|---|---|---|---|
| *Data collection and processing* | | | | |
| Magnification | 60,000 | 60,000 | 60,000 | 60,000 |
| Voltage (kV) | 300 | 300 | 300 | 300 |
| Electron exposure (e$^-$/Å$^2$) | 60 | 60 | 60 | 60 |
| Defocus range (μm) | −0.5 to −2.0 | −0.5 to −2.0 | −0.5 to −2.0 | −0.5 to −2.0 |
| Pixel size (Å) | 0.870 | 0.870 | 0.874 | 0.874 |
| Symmetry imposed | C1 | C3 | C1 | C1 |
| Imported movies (no.) | 4175 | 4175 | 2124 | 2124 |
| Initial particle images (no.) | 878,975 | 878,975 | 766,652 | 766,652 |
| Final particle images (no.) | 38,004 | 44,292 | 67,529 | 48,715 |
| Map resolution (Å) | 3.03 | 2.70 | 3.20 | 3.29 |
| FSC threshold | 0.143 | 0.143 | 0.143 | 0.143 |
| *Refinement* | | | | |
| Model vs. Map CC (volume) | 0.79 | | | |
| Protein residues | 3624 | | | |
| R.m.s. deviations | | | | |
| Bond lengths (Å) | 0.003 | | | |
| Bond angles (°) | 0.516 | | | |
| Validation | | | | |
| MolProbity score | 1.81 | | | |
| Clashscore | 9.09 | | | |
| Rotamer outliers (%) | 0.03 | | | |
| Ramachandran plot | | | | |
| Favoured (%) | 95.30 | | | |
| Allowed (%) | 4.61 | | | |
| Disallowed (%) | 0.08 | | | |

CoV-2 panned sublibraries. We synthesized the top read clones in each family as linked dimers. Among them, four (C17, C49, C116 and C246) out of eight and five (P17, P86, P158, P334 and P543) out of ten dimer clones were expressed and further analysed.

**Nanobody expression**. Each selected amino acid sequence was connected with a (GGGGS)$_4$ linker as a tandem dimer; coding genes of these and of the previously reported SARS72 dimer, mNb6 dimer and Ty1 monomer were codon-optimized and synthesized (Eurofins Genomics Inc., Tokyo, Japan). The synthesized genes were subcloned in the pMES4 vector to express N-terminal PelB signal peptide-conjugated and C-terminal 6×His-tagged or Flag-tagged nanobodies into the bacterial periplasm. These gene constructs were transformed into BL21(DE3) *E. coli* cells (BioDynamics Laboratory Inc., Tokyo, Japan) and plated on LB agar with ampicillin, which were incubated at 37 °C overnight. Colonies were picked and cultured at 37 °C to reach an OD of 0.6 AU; the cells were cultured at 37 °C for 3 h or at 28 °C overnight with 1 mM IPTG (isopropyl-β-D-thiogalactopyranoside: Nacalai). Cultured cells were collected by centrifugation. Nanobodies were eluted from the periplasm by soaking in a high osmotic buffer containing 200 mM Tris, 0.5 mM EDTA and 500 mM sucrose (pH 8.0) at 4 °C for 1 h. They were incubated with 2× volumes of a diluted buffer containing 50 mM Tris, 0.125 mM EDTA, and 125 mM sucrose (pH 8.0) with a trace amount of benzonase nuclease (Merck KGaA, Darmstadt, Germany) at 4 °C for 45 min, and the supernatants were centrifuged (20,000 × *g*, 4 °C for 10 min). The supernatants were sterilized with the addition of gentamicin (Thermo) and passed through a 0.22-μm filter (Sartorius AG, Göttingen, Germany). The filtered supernatants of Flag-tagged trimer were used for ELISA and His-tagged dimers were applied to a HisTrap HP nickel column (Cytiva) equipped on an ÄKTA purifier HPLC system (Cytiva) and washed; the bound His-tagged protein was eluted with 300 mM imidazole. The elution was concentrated with a VIVAspin 3000-molecular weight cut off (MWCO) filter column (Sartorius) and applied to a Superdex75increase 10/300 GL gel-filtration column (Cytiva) equipped on an ÄKTA pure HPLC system (Cytiva) to obtain the dimer fractions and exclude cleaved monomers and imidazole. Purity was measured via Coomassie Brilliant Blue (CBB) staining.

**Antibodies**. Antibodies used for western blotting and cell staining were anti-His (rabbit polyclonal PM032: Medical and Biological Laboratories Co., Ltd. (MBL), Nagoya, Japan), anti-His (rabbit monoclonal EPR20547: ab213204, Abcam), anti-Flag (mouse monoclonal M2: Thermo) and anti-rhodopsin C9 (TETSQVAPA) (mouse monoclonal 1D4, sc-57432: Santa Cruz Biotechnology Inc., CA). Horseradish peroxidase (HRP)-linked secondary antibodies included anti-mouse IgG (sheep polyclonal, NA931: GE Healthcare, Buckinghamshire, UK), anti-rabbit IgG (sheep polyclonal, NA934: GE Healthcare), anti-alpaca IgG[85] (goat polyclonal, 128-035-232: Jackson ImmunoResearch Laboratories, Inc., West Grove, PA), Alexa

Fluor 488 goat anti-mouse IgG (rabbit polyclonal, P0449: Dako, Glostrup, Denmark) and Alexa Fluor 594 goat anti-rabbit IgG (Dako).

**Nasal swab specimens and western blotting**. Nasal swab specimens were collected on 1 February 2022 (I and J), 3 February 2022 (K to P), 10 February 2022 (A to H) and 23 March 2022 (Q to V) from emergency fever patients. All specimens were tested to be SARS-CoV-2 positive by PCR. Sixteen samples (A to D, I to R, T and V) were confirmed to be SARS-CoV-2 positive and 6 samples (E to H, S and U) were not confirmed to be positive by the PCR test.

Samples were incubated at 37 °C for 30 min (for SARS-CoV-2 spike variants and swab specimens) or boiled for 2 min (for nanobodies) with Laemmli's SDS sample buffer containing 2.5 mM Tris (pH 6.8), 2% SDS, 10% glycerol, 0.001% bromophenol blue and 13.3 mM dithiothreitol (DTT). Samples were electrophoretically separated on 5–20% or 15–25% gradient polyacrylamide gels and electrophoretically transferred onto polyvinylidene difluoride (PVDF) membranes (Immobilon-P: Millipore, Billerica, MA). Blotted membranes were incubated overnight at 4 °C with the C9 antibody (the dilution ratio was 1:5000) or the C-terminally 6×His-tagged homodimer of nanobodies in Tris-buffered saline (TBS, pH 7.4) containing 0.005% Tween 20 (TBST) and 5% skim milk—the concentrations of P158, P334 and P543 were 0.2, 1.0 and 0.4 μg ml$^{-1}$, respectively. For testing them for the Omicron BA.1 variant, concentrations of P86 and C246 were 0.4 μg ml$^{-1}$, and of the other nanobodies (C17, P17, C49 and C116) were 5.0 μg ml$^{-1}$. In the case of nanobody-based blotting, after 3 washes with TBST, the membranes were incubated with 1:5000-diluted anti-His-tag antibody (MBL) in TBST containing 5% skim milk at room temperature for 1 h. The membranes were soaked with 1:5000-diluted HRP-conjugated anti-rabbit or anti-mouse IgG secondary antibodies (GE Healthcare) in TBST containing 5% skim milk for 30 min at room temperature. After 3 washes with TBST, reactive protein bands were visualized using an ECL Plus system (Cytiva). For CBB R-250 staining, a Rapid Stain CBB Kit (Nacalai) was used according to the manufacturer's protocol.

**Column chromatography for protein purification**. HEK293T cells expressing the SARS-CoV-2 S2 domain or the extracellular domain of the HCoV-OC43 spike were cultured in serum-free Opti-MEM (modified Eagle's medium: Thermo) containing 1% penicillin and streptomycin. After 48 h, the culture supernatants were centrifuged, filtered, and concentrated with VIVAspin 20 size exclusion columns (30,000-MWCO). Pellets of HEK293T cells expressing the extracellular domain of the SARS-CoV-2 spike were suspended in 50 mM phosphate buffer (pH 7.4) containing 1% DDM, 0.1% CHAPS, 0.001% CHS, 0.1% LMNG and 500 mM NaCl. The suspension was passed through a 21-gauge needle (Terumo Co., Tokyo, Japan) several times, trace amounts of benzonase nuclease (Merck) were added, and the lysates were incubated at 4 °C overnight. The cell lysate was cleared by centrifugation and filtration. The C-terminal 6×His-tagged spike protein was

## Table 2 Crystallographic data collection and refinement statistics.

| | |
|---|---|
| *Data collection* | |
| X-ray source | SPring-8 BL44XU |
| Wavelength (Å) | 0.9000 |
| Space group | *C*2 |
| Unit cell *a, b, c* (Å) | 105.08, 43.17, 56.03 |
| Unit cell *α, β, γ* (°) | 90.00, 110.76, 90.00 |
| Resolution range (Å) | 49.13–1.60 (1.63–1.60)[a] |
| Total no. of reflections/no. of unique reflections | 103,494 (5156)/ 30,851 (1493) |
| Completeness (%) | 99.1 (99.5) |
| Redundancy | 3.4 (3.5) |
| $\langle I/\sigma(I)\rangle$ | 11.9 (2.8) |
| $R_{meas}$ (all I+ & I−) | 0.067 (0.668) |
| $R_{meas}$ (within I+/I−) | 0.065 (0.628) |
| $CC_{1/2}$ | 0.998 (0.835) |
| *Refinement* | |
| Resolution range (Å) | 30.80–1.60 (1.65–1.60) |
| Completeness (%) | 98.8 (99.0) |
| No. of reflections, working set | 30,833 (3054) |
| No. of reflections, test set | 1510 (138) |
| $R_{work}/R_{free}$ | 0.177/0.213 (0.236/0.245) |
| No. of non-H atoms | 2167 |
| Protein | 1885 |
| Ion/Ligand | 54 |
| Water | 232 |
| R.m.s. deviation bonds (Å), angles (°) | 0.006, 0.87 |
| Average *B* factors (Å²) | 23.61 |
| Protein | 22.29 |
| Ion/Ligand | 34.89 |
| Water | 31.84 |
| Ramachandran favoured/allowed/ disallowed (%) | 98.25/1.75/0 |
| PDB code ID | 7VPY |

[a]Values in parentheses refer to the highest resolution shell.

purified through a HisTrap HP 1 ml column equipped on an ÄKTA pure HPLC system (Cytiva) under step-by-step elution conditions using running (20 mM imidazole) and eluting (500 mM imidazole) phosphate buffers containing 20 mM phosphate, 500 mM NaCl, 0.5% DDM, 0.1% CHAPS, 0.001% CHS, and 0.1% LMNG (for spike proteins from the cell lysate) (pH 7.4). Elution fractions of 6×His-tagged proteins were identified via western blotting. Anti-6×His-tagged antibody-positive fractions were gathered and concentrated via VIVAspin 6 size exclusion columns (30,000-MWCO) to reach a volume under 0.5 ml. Size exclusion chromatography experiments were performed on a Superose6increase 10/300 GL column (Cytiva) with an ÄKTA pure HPLC system (Cytiva). Sample volumes were ~0.5 ml; injected samples were separated in PBS or PBS-LMNG (for SARS-CoV-2 spike protein) in a chilled chamber with a flow rate of 0.1 ml min⁻¹. Chromatograms were monitored at 280 nm using a UV spectrophotometer. Elution fractions were identified via western blotting, gathered, and concentrated via VIVAspin 6 size exclusion columns (30,000-MWCO). Protein concentrations were measured with a NanoDrop 2000c spectrophotometer (1 mAU at 280 nm was equivalent to 1 mg ml⁻¹) (Thermo).

**Cell culture and transfection**. HEK293T, K562 and HOS (human osteosarcoma) cells were grown in Dulbecco's modified Eagle's medium (DMEM: Invitrogen) supplemented with 10% foetal bovine serum (FBS) and antibiotics (1% penicillin and streptomycin). The cells were cultured in a humidified incubator with 5% CO₂ at 37 °C. The human ACE2 and TMPRSS2 genes were transduced into K562 and HOS cells with a lentivirus. The lentiviral vector pWPI-IRES-Bla-Ak-ACE2-TMPRSS2 was acquired from AddGene (plasmid #154983). DT40 cells (from the Japanese Collection of Research Bioresources Cell Bank, Osaka, Japan) were grown in RPMI1640 medium (Invitrogen) supplemented with 10% FBS, 10% chicken serum, 1% penicillin and 1% streptomycin.

**Measuring titres and nanobody-based sandwich ELISA**. Two micrograms of the recombinant extracellular domain of the SARS-CoV-2 spike or 10 µg of the homodimer of nanobodies was diluted with 10 ml of 0.1 M carbohydrate buffer (pH 9.8). Each well of a MaxiSorp 96-well plate (Thermo) was coated with 100 µl of the diluent at 4 °C overnight. To measure the titres of the immunized two alpacas,

the wells were washed 3 times with high-salt PBS-LMNG (500 mM NaCl and 0.001% LMNG) and blocked with the high-salt PBS-LMNG containing 1% FBS at room temperature for 1 h. For screening conditions with the nanobody-based sandwich ELISA for the SARS-CoV-2 spike, the wells were washed with PBST (0.05% Tween 20, pH 7.4) and were blocked at room temperature for 1 h with five kinds of blocking solutions: 1×Carbo-Free blocking solution (Vector Laboratories, Inc., Burlingame, CA); 5% bovine serum albumin (BSA) (Sigma-Aldrich) in PBST; 5% skim milk in PBST; 3% casein (Merck) in 20 mM TBS (pH 11.0); or 5% polyvinylpyrrolidone (PVP) average molecular weight 10,000 (Sigma-Aldrich) in PBST. For measuring titres, 10 µl of a 0.1% dilution of serum from immunized alpacas in PBST was added to the well and incubated at room temperature for 1 h. After 3 washes with PBST, HRP-conjugated anti-alpaca VHH antibody (Jackson) at a dilution of 1:5000 was reacted at room temperature for 30 min. For screening of nanobody-based sandwich ELISA conditions, 100 µl of 1% (w/v) of the extracellular domain of SARS-CoV-2 spike (1 µg 100 µl⁻¹) in high-salt PBS-LMNG was added to each well and captured for 1 h at room temperature for 1 h. The well was washed 3 times with the high-salt PBS-LMNG; each 30 ng of biotin-conjugated nanobody in 100 µl (0.03% w/v) of PBS-LMNG was soaked at room temperature for 30 min. After 3 washes with PBS-LMNG, 100 µl of HRP-conjugated streptavidin (TCI) at a dilution of 1:5000 in PBS-LMNG was incubated at room temperature for 30 min.

The amounts of remaining HRP conjugates after 3 washes with the buffer were measured with the addition of 100 µl of 50 mM phosphate-citrate buffer (pH 5.0) containing 0.4 µg of *o*-phenylenediamine dihydrochloride (OPD) (Sigma-Aldrich) to develop at room temperature for 30 min, after which the reaction was stopped with the addition of 10 µl of 5 M sulfuric acid (H₂SO₄). Each well was read at an optical density (OD) of 450 nm using a microplate reader (Bio-Rad).

Each well of a MaxiSorp 96-well plate was coated with 100 µl of the diluent containing 10 µg of P158 diluted with 10 ml of 0.1 M carbohydrate buffer, pH 9.8. The P158-coated wells were blocked with 3% BSA in PBST. Nasal swab specimens were dissolved with 1% DDM and benzonase nuclease. Lysed samples in PBS were diluted with water to contain the final 20 mM NaCl concentrations. Dissolved and diluted nasal swab samples containing 1% BSA were added to the wells. To increase sensitivity, the Flag-tagged P86 trimer nanobody was used as a primary antibody at the concentration of 30 ng 100 µl⁻¹; the 1:5000-diluted anti-Flag antibody (secondary antibody) and the 1:5000-diluted HRP-conjugated anti-mouse antibody (final antibody) were used for detection.

**Immunochromatography**. Antigen test kits detecting the SARS-CoV-2 spike based on nitrocellulose lateral flow assays were developed by Yamato Scientific Co., Ltd. (Tokyo, Japan). Briefly, purified P158 (330 ng µl⁻¹) was lined ~1 mm wide on an IAB90 nitrocellulose membrane (Advantech, Toyo Roshi Kaisha, Ltd., Tokyo, Japan). The membrane was soaked in 1×Carbo-Free blocking solution at room temperature for 1 h and air-dried. Purified P86 or P543 nanobodies (500 ng µl⁻¹) were amine coupled to 30-nm diameter Estapore beads (Sigma-Aldrich) according to the manufacturer's protocol. Glass fibre paper (GF/DVA: Cytiva) was soaked in 1×Carbo-Free blocking solution containing 0.005% (w/v) nanobody-coupled Estapore beads for saturation and then dried under vacuum conditions. The lined nitrocellulose membrane and the bead-absorbed glass fibre paper were overlaid on a backing sheet (Cytiva). CF4 paper (Cytiva) was used for both the sample pad and the absorbance pad; they were overlaid on the prepared backing sheet as well. The four-layered sheet was cut 5 mm wide and housed in black cases: the cassettes were stored in sealed packages with silica gels. When dried, 150 µl of sample was spotted onto the sample pad; the kits were photographed under a 315 nm UV lamp for an arbitrary amount of time.

Antigen test kits detecting the SARS-CoV-2 spike from nasal swab specimens have been further modified. Purified P158 (330 ng µl⁻¹) was lined ~2 mm wide on the IAB90 nitrocellulose membrane; The membrane was soaked in 3% BSA blocking solution at room temperature for 1 h and air-dried; Purified P543 trimer (500 ng µl⁻¹) was amine coupled to 30-nm diameter Estapore beads. Glass fibre paper was soaked in 3% BSA blocking solution containing 0.005% (w/v) nanobody-coupled Estapore beads for saturation and then dried under vacuum conditions. The nasal swab specimens were sampled with 1% DDM and nuclease and were diluted with water to contain the final 20 mM NaCl concentration. Approximately, 120 µl of the sample was spotted on the sample pad; the strips were photographed under a 385 nm UV lamp.

**Kinetic assays via biolayer interferometry (BLI)**. Real-time binding experiments were performed using an Octet Red96 instrument (fortèBIO, Pall Life Science, Portsmouth, NH). Each purified nanobody clone was biotinylated with EZ-Link Sulfo-NHS-LC-Biotin (Thermo) according to the manufacturer's protocol; uncoupled biotin was excluded with a size exclusion spin column (PD SpinTrap G-25: Cytiva) in PBS (pH 7.4). Assays were performed at 30 °C with shaking at 1000 rpm. Biotin-conjugated clones at 10 µg ml⁻¹ were captured on a streptavidin-coated sensor chip (SA: fortèBIO) to reach the signals at 4 nm. One uncoated sensor chip was monitored as the baseline; another biotin-conjugated anti-IL-6-coated sensor chip (anti-IL-6 nanobody: COGNANO Inc.) was monitored as the background. The remaining 6 channels were immobilized with biotinylated anti-SARS-CoV-2 spike clones, and real-time binding kinetics to the purified extracellular domain of the SARS-CoV-2 spike trimer complex were measured in sequentially diluted concentrations at the same time (8 channels per assay). The

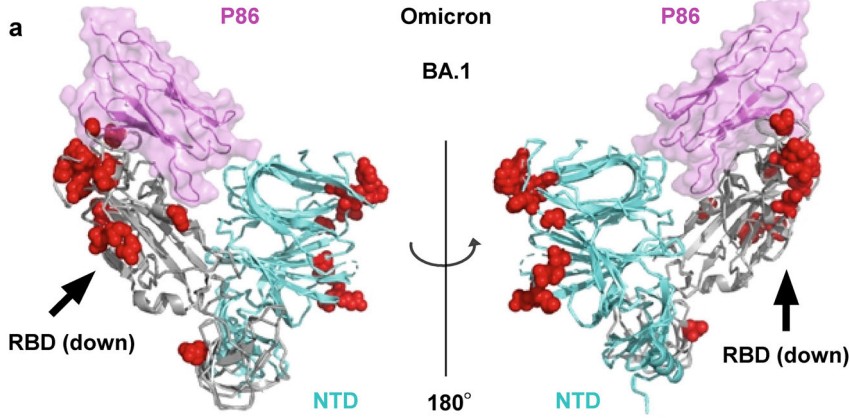

A67V, H69-V70del, T95I, G142D, V143-Y145del, N211I, L212del, R214EPE, G339D, S371L, S373P, S375F, K417N, N440K, G446S, S477N, T478K, E484A, Q493R, G496S, Q498R, N501Y, Y505H, T547K

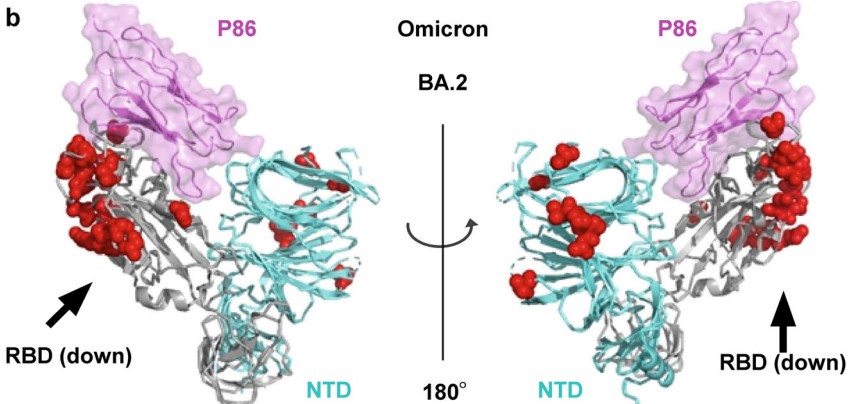

T19I, L24S, P25-P26del, A27S, G142D, V213G, G339D, S371F, S373P, S375F, T376A, D405N, R408S, K417N, N440K, S477N, T478K, E484A, Q493R, Q498R, N501Y, Y505H

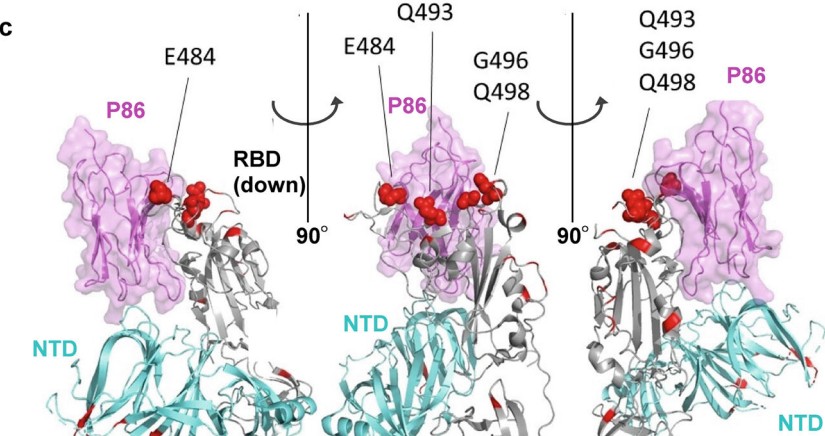

**Fig. 7 Fitting the model of P86 with the SARS-CoV-2 Omicron variants. a**, **b** The down-RBD, the neighbouring NTD, and P86 are coloured grey, cyan, and pink, respectively. Spike protein mutations of the SARS-CoV-2 Omicron variants BA.1 (**a**, **c**) and BA.2 (**b**) are depicted in red. **c** A close-up view of the binding domain of P86 with the RBD denoting residues E484, Q493, G496 and Q498.

concentrations of the SARS-CoV-2 spike loaded varied between 1 and 32 µg ml⁻¹, corresponding to 1–32 nM or less, when an average of the molecular weight of the SARS-CoV-2 spike trimer complex was estimated to be ~1000 kDa or more according to chromatograms of gel filtration column chromatography (see Supplementary Fig. 1b). Assays were performed with high-salt phosphate buffer containing 500 mM NaCl and 0.001% LMNG (stabilized spike: for P17, P86, P334 and C116) or with hypotonic phosphate buffer containing 25 mM NaCl and 0.00005% LMNG (fluctuated spike: for P158, P543, C17, C49 and C246). After baseline equilibration for 180 s in each buffer, the association and dissociation were carried out for 600 s each. The data were double subtracted before fitting was performed with a 1:1 fitting model in fortèBIO data analysis software. The equilibrium dissociation constant ($K_D$), $k_{off}$ and $k_{on}$ values were determined with a global fit applied to all data.

**Pseudotyped virus production**. HIV-1-based SARS-CoV-2 spike pseudotyped virus was prepared as follows: LentiX-HEK293T cells were transfected using a polyethyleneimine transfection reagent (Cytiva) with plasmids encoding the C-terminally C9-tagged full-length SARS-CoV-2 spike variants (original, Alpha, Beta, Delta and Omicron) and HIV-1 transfer vector encoding a luciferase reporter, according to the manufacturer's protocol. The mutations of each variant are as follows: original (D614G); Alpha (H69del, V70del, Y144del, N501Y, A570D, D614G, P681H, T716I, S982A and D1118H); Beta (L18F, D80A, D215G, R246I, K417N, E484K, N501Y, D614G and A701Y); Delta (T19R, G142D, E156del, F157del, R158G, L452R, T478K, D614G, P681R and D950N); and Omicron (A67V, H69del, V70del, T95I, G142D, V143-Y145del, N211I, L212del, R214EPE, G339D, S371L, S373P, K417N, N440K, G446S, S477N, T478K, E484A, Q493R, G496S, Q498R, N501Y, Y505H, T547K, D614G, H655Y, N679K, P681H, N764K, D796Y,

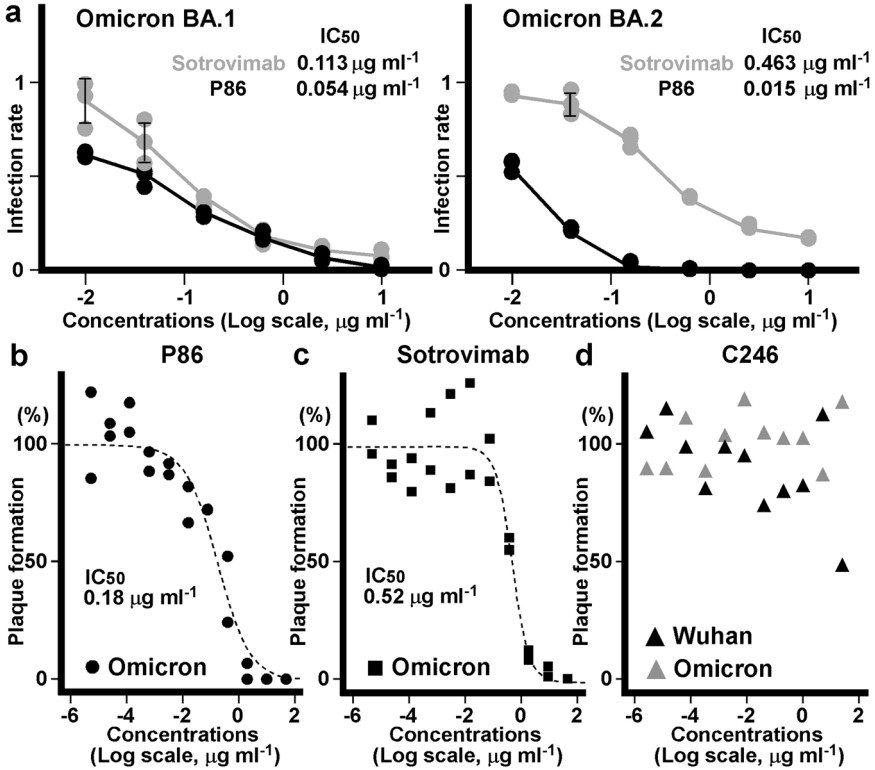

**Fig. 8 Pseudotyped and authentic virus neutralization assays. a** Means of the number of infectious units of virus for the SARS-CoV-2 spike Omicron variants ($n = 3$). Human osteosarcoma (HOS) cells expressing ACE2 and TMPRSS2 infected with HIV-based pseudotyped virus were assessed. Relative infection units at the indicated concentrations of the P86 (black) and sotrovimab (grey) are plotted; means ± standard deviation (SD) are shown. Measured IC$_{50}$s for the SARS-CoV-2 Omicron variants BA.1 and BA.2 of P86 and sotrovimab are inset. **b, c** Concentration dependency of the P86 nanobody (**b**) and sotrovimab antibody (**c**) for plaque reduction of the authentic SARS-CoV-2 Omicron BA.1 variant virus. The ratios of plaque formation are expressed as a percentage of the plaque numbers from untreated cells and fitted with a sigmoidal function using Igor Pro (ver. 8.04, Wavemetrics). Data are representative of two independent experiments. **d** The C246 nanobody versus plaque formation of the authentic Wuhan and Omicron BA.1 variant viruses. The ratios of plaque formation are shown as a percentage of the plaque numbers from untreated cells ($n = 1$).

N856K, Q954H, N969K and L981F)[86]. Cells were incubated for 4–6 h at 37 °C with the medium that was then replaced with DMEM containing 10% FBS for the following 48-h culture. The supernatants were then harvested, filtered through a 0.45-μm membrane, concentrated with ultracentrifugation and frozen at −80 °C.

**Pseudotyped virus neutralization assay.** Fivefold sequentially diluted nanobodies (from 10 μg ml$^{-1}$) were incubated with SARS-CoV-2 pseudotyped viruses for 1 h. K562 and HOS cells expressing human ACE2 and TMPRSS2 were subsequently infected with the antibody-virus mixture for 1 h at 37 °C and cultured for 2 days. The cells were lysed, and luciferase activity was measured using the Steady-Glo Luciferase Assay System (Promega KK, Osaka, Japan) with a microplate spectrophotometer (ARVO X3: PerkinElmer Japan Co., Ltd., Kanagawa, Japan). The obtained relative luminescence units were normalized to those derived from cells infected with the SARS-CoV-2 pseudotyped virus in the absence of nanobodies.

**Flow cytometry.** The ability of nanobodies to bind to the cell surface of the SARS-CoV-2 spike was studied by fluorescence-activated cell sorting (FACS)[87]. K562 cells expressing the SARS-CoV-2 spike were incubated with 1 μg ml$^{-1}$ purified nanobody on ice for 30 min. After washing, the cells were incubated with the 1:400-diluted anti-His antibody (Abcam) on ice for 30 min and then the 1:400-diluted Alexa 647-conjugated anti-rabbit IgG (Dako). The cells were analysed with a Beckman-Coulter FC-500 Analyzer (Coulter Electronics, Hialeah, FL). The Ty1 and B9[88] nanobodies were used as controls. A region of positive signals for Ty1 was square-gated.

DT40 cells stably expressing the SARS-CoV-2 Omicron BA.1 variant spike were incubated with 1 μg ml$^{-1}$ purified nanobodies and the 1:400-diluted antibodies. The cells were analysed with the BD Accuri flow cytometer (Becton, Dickinson and Company, Franklin Lakes, NJ). A homemade anti-Her-2 nanobody (COGNANO Inc.) was used as a control.

**Microscopy analyses for cell staining.** HEK293T cells were transiently transfected with plasmids encoding the C-terminally C9-tagged full-length SARS-CoV-2 spike variants using Lipofectamine 3000 (Thermo) according to the manufacturer's

instructions. The next day, the cells were seeded on collagen type I-coated culture plates (IWAKI, AGC TECHNO GLASS CO., LTD., Shizuoka, Japan) and cultured for 24 h before being fixed with 2% paraformaldehyde (PFA) at 4 °C overnight. After 3 washes with PBST (0.005% Tween), the cells were blocked with PBST containing 2% goat serum (blocking solution) at room temperature for 1 h. Each well was soaked with 100 μl of the blocking solution containing 30 ng of purified nanobody, except for 6 ng of C116, at 4 °C overnight. After washing with PBST, the 1:400-diluted anti-His-tagged antibody and the 1:400-diluted anti-C9-tagged antibody in blocking buffer were added and reacted at room temperature for 1 h. Finally, after washing, the fluorescently conjugated anti-rabbit IgG (594 nm emission) and anti-mouse IgG (488 nm emission) antibodies (Alexa Fluor: Thermo) were diluted with blocking buffer (1:400) and added to the wells, and the fixed cells were labelled at room temperature for 1 h before washing 3 times with PBST. Cellular nuclei were visualized with 4′,6-diamidino-2-phenylindole (DAPI). Stained cells were imaged with a 2-ms exposure time (594 nm emission), with a 10-ms exposure time (488 nm emission), or automatically adjusted exposure time (DAPI) using microscopy (IX71S1F-3: Olympus Corporation, Tokyo, Japan) with the cellSens Standard 1.11 application (Olympus). A $3 \times 4$ cm$^2$ printed rectangle corresponds to a $165 \times 220$ μm$^2$ observed field.

**Expression and purification of monomeric P86 and P17.** The monomeric C-terminally 6×His tagged nanobody genes were cloned into the pMES4 vector. The complete amino acid sequences are as follows (signal peptide sequence is underlined). P86: <u>MKYLLPTAAAGLLLLAAQPAMA</u>QVQLQESGGGLVQAGGS LRLSCVASGRTFSSLNIVWFRQAPGKERKFVAAINDRNTAYAESVKGRFTISRD NAKNTVHLQMNSLKPEDTAVYYCHSADVNGGMDYWGKGTQVTVSSHHH HHH. P17: <u>MKYLLPTAAAGLLLLAAQPAMA</u>QVQLQESGGGLVQAGGSLRLS CAASGRTS<u>SVYNMAWFRQTPGKEREFVAAIT</u>GNGGTTLYADSVKGRLTISRG NAKNTVSLQMNVLKPDDTAVYYCAAGGWGKERNYAYWGQGTQVTVSSH HHHHH.

Bacterial BL21(DE3) *E. coli* cells were transformed with the plasmids and grown on an LB ampicillin-supplemented plate; colonies were picked and inoculated into 5 ml of LB medium containing 200 μg ml$^{-1}$ ampicillin; and the cells were cultured in a shaking incubator overnight at 37 °C. The cultures were transferred to 1 L of

LB medium containing 200 μg ml⁻¹ ampicillin. At an optical density below 0.6, cells were cultured for 3 h at 37 °C with 1 mM IPTG.

The bacterial pellet was collected by centrifuging at 6000 × *g* for 30 min and suspended in 45 ml of lysis buffer containing 20 mM Tris (pH 8.0), 0.68 mM EDTA, 500 mM sucrose and a trace amount of benzonase nuclease (Merck). The solutions were mixed on a TR-118 tube rotator (AS ONE Corporation, Osaka, Japan); 90 mL of 20 mM Tris (pH 8.0) was added, and the mixtures were rotated for 45 min at 4 °C. The resulting solutions were centrifuged at 20,000 × *g* for 10 min at 4 °C. The supernatant was filtered and loaded onto a Ni-NTA column (Cytiva) equilibrated with 20 mM Tris buffer (pH 8.0). The column was washed several times with 20 mM Tris buffer (pH 8.0) containing 40 mM imidazole for P86 or 20 mM Tris buffer (pH 8.0) containing 150 mM NaCl and 40 mM imidazole for P17. Then, the nanobody was eluted with 20 mM Tris buffer (pH 8.0) containing 250 mM imidazole for P86 or 150 mM NaCl and 350 mM imidazole for P17.

The fractions containing nanobodies were collected and further purified using a HiLoad 16/60 Superdex 75 gel filtration column (Cytiva) equilibrated with 20 mM Tris buffer (pH 8.0) containing 150 mM NaCl for P86 or with 20 mM HEPES buffer (pH 8.0) containing 150 mM NaCl for P17. The peak fractions were collected and concentrated. The final concentrations of the P86 and P17 sample were 51 and 3.0 mg ml⁻¹, respectively, based on the use of the absorption coefficient at 280 nm.

**Cryo-EM specimen preparation and data collection.** An epoxidized graphene grid (EG-grid)[89] was used to increase the number of protein particles. The trimer complex of the SARS-CoV-2 spike (D614G) with the furin-resistant mutation ("GSAS") at a concentration of 0.1 mg ml⁻¹ was mixed with a 5-time molar excess of P86 or P17 monomer and incubated on ice for 10 min, and 3 μl of the spike-nanobody complex solution was applied to the EG-grid. After incubation at room temperature for 5 min, the grids were blotted with a force of –3 and a time of 2 s in a Vitrobot Mark IV chamber (Thermo) equilibrated at 4 °C and 100% humidity and then immediately plunged into liquid ethane. Excessive ethane was removed with filter paper, and the grids were stored in liquid nitrogen. All cryo-EM image datasets were acquired using a JEM-Z300FSC (CRYO ARM 300: JEOL, Tokyo, Japan) operated at 300 kV with a K3 direct electron detector (Gatan, Inc., Pleasanton, CA) in CDS mode[90]. The Ω-type in-column energy filter was operated with a slit width of 20 eV for zero-loss imaging. The nominal magnification was ×60,000, corresponding to 0.870 Å per pixel. Defocus varied between –0.5 and –2.0 μm. Each movie was fractionated into 60 frames (0.0505 s each, total exposure: 3.04 s) with a total dose of 60 e⁻/Å².

**Cryo-EM image processing and refinement.** The images were processed using RELION 3.1[91]. Movies were motion corrected using MotionCor2[92], and the contrast transfer functions (CTFs) were estimated using CTFFIND 4.1[93]. Micrographs whose CTF max resolutions were beyond 5 Å were selected. Three-dimensional (3D) template-based autopicking was performed for all images, and the particles were extracted with 4× binning, which were subjected to two rounds of 2D classification. An initial model was generated and used as a reference for the following 3D classification in the P86 dataset. In the P17 dataset, a density map of spike trimers (our previous dataset) was used as a reference. Reference-based 3D classification (into 4 classes) without applying symmetry was conducted, and the selected particles were re-extracted without binning. 3D autorefinement without applying symmetry, soft mask generation, postprocessing, CTF refinement, Bayesian polishing, and another round of 3D autorefinement were performed. Then, focused 3D classification without alignment was performed to separate up- and down-states in one RBD. The selected particles were subjected to another round of 3D autorefinement, postprocessing, CTF refinement, 3D autorefinement. C3 symmetry was applied during the final round of 3D autorefinement for the 3-up +P86 dataset. The data were imported and further processed with non-uniform refinement in cryoSPARC v3.2.0[94]. The final map resolutions (FSC = 0.143) were 3.03 and 2.70 Å in the 2-up and 3-up states in the P86 dataset and 3.20 and 3.29 Å in the 1-up and 2-up states in the P17 dataset, respectively. For the 2-up+P86 dataset, we tried local refinement to visualize the density of P86 bound to the down-RBD, but the resolution was limited to 5.13 Å (FSC = 0.143).

The model was built for the 2-up+P86 dataset. The SARS-CoV-2 spike trimer D614G mutant (PDB entry: 7KRR)[95] and the crystal structure of P86 were used initial models. After the models were manually fitted into the density using UCSF Chimera v1.15[96] and modified in Coot v0.8.9.2[97], real-space refinement was performed in PHENIX v1.19.1[98]. The model was validated using MolProbity[99], and this cycle was repeated several times. Figures were prepared using UCSF Chimera[96], ChimeraX[100] and PyMOL v2.5.0 (Schrödinger, LLC, New York, NY). The parameters are summarized in Table 1.

**Crystallization of P86.** Crystallization was performed by the sitting drop vapour diffusion method. A mosquito crystallization machine (TTP LabTech Ltd., Melbourn, Hertfordshire, UK) was used to prepare drops on 96-well VIOLAMO plates (AS ONE). The reservoir solution was 60 μl in volume, and 0.1 μl of protein solution was mixed with 0.1 μl of reservoir solution. Crystals appeared under the condition of 51 mg ml⁻¹ protein, 0.2 M ammonium sulfate, and 30% (w/v) polyethylene glycol 4000 at 20 °C. A crystal cluster was crushed, and a peeled single

crystal was harvested by LithoLoop (Protein Wave, Nara, Japan). Before the crystal was frozen in liquid nitrogen, it was soaked in the crystallization solution supplemented with 5% (v/v) ethylene glycol.

**X-ray data collection, processing, structure solution and refinement.** An X-ray diffraction experiment was performed on the BL44XU beamline of SPring-8 (Hyogo, Japan). Diffraction images were collected at 100 K using an EIGER X 16 M detector (Dectris, Philadelphia, PA, USA). The beam size was 50.0 × 50.0 μm² (h × w). A 0.8-mm Al attenuator was used to weaken the X-ray. The crystal-to-detector distance was 160 mm. The exposure time per frame and the oscillation angle were 0.1 s and 0.1°, respectively. A total of 1800 images were collected. The dataset was processed using XDS[101] and scaled by Aimless[102]. Molecular replacement phase determination was performed by MOLREP[103] with a nanobody structure (PDB code ID: 5IVO)[104] as a search model. Initial model building was performed by PhenixAutoBuild implemented in PHENIX[98]. Manual model building was performed using Coot[97]. The programme refmac5[105] in the ccp4 suite[106] and the programme Phenix-refine[98] were used for structural refinement. The stereochemical quality of the final model was checked by MolProbity[99]. Data collection and refinement statistics are summarized in Table 2.

**Authentic virus neutralization assay.** TMPRSS2-expressing Vero E6 (VeroE6 + TMPRSS2) cells (JCRB1819)[107] were maintained in high-glucose DMEM containing 10% FBS, 1% penicillin and streptomycin and 1 mg ml⁻¹ G418 (Nacalai).

All experiments with authentic SARS-CoV-2 viruses were performed in a biosafety level-3 (BSL-3) facility at the Research Institute for Microbial Diseases, Osaka University. A Wuhan variant (strain SARS-CoV-2/Hu/DP/Kng/19-020) was kindly provided by Dr. Sakuragi at the Kanagawa Prefectural Institute of Public Health. An Omicron variant BA.1 (strain hCoV-19/Japan/TY38-873/2021) was obtained from the National Institute of Infectious Diseases, Japan. All viruses were initially amplified in VeroE6 + TMPRSS2 cells, and the culture supernatants were harvested and stored at −80 °C until use. The infectious virus titre was determined as plaque-forming units (PFU) by plaque assay.

Antibodies and viruses were diluted with DMEM containing 1% penicillin and streptomycin, and 2% FBS. Serially diluted nanobodies (from 50 μg ml⁻¹ for P86 and 25 μg ml⁻¹ for C246) and antibodies (from 50 μg ml⁻¹) were mixed with an equal volume of virus solution containing 100 PFU. After incubation for an hour, the mixture was inoculated into confluent VeroE6 + TMPRSS2 cells in a 6-well plate for 1 h at 37 °C with 5% CO₂. After washing the cells with DMEM containing 2% FBS, the cells were overlaid with 2 mL of DMEM containing 1% agarose, 5% FBS, 10 mM HEPES (Invitrogen), 0.3% sodium hydrogen carbonate (Invitrogen). At a few days post-infection, 10% formalin neutral buffer solution (FUJIFILM Wako Pure Chemical Co., Osaka, Japan) was overlaid to fix the cells. After removing the formalin and solid overlay medium, the fixed cells were stained with 0.1% crystal violet (bioWORLD, Dublin, OH), and then the number of plaques was counted.

**Statistics and reproducibility.** The number of replicates is indicated in the figure legend and means ± standard deviation (SD) are shown in graphs.

**Reporting summary.** Further information on research design is available in the Nature Research Reporting Summary linked to this article.

## Data availability

Density maps are available at the Electron microscopy Data Bank (EMDB) and Protein Data Bank (PDB) with accession codes EMD-32078 (2-up+P86), EMD-32079 (3-up +P86), EMD-32080 (1-up+P17), EMD-32081 (2-up+P17) and PDB-7VQ0 (2-up+P86). Additional cryo-EM data supporting this study are available from Ke.N. on reasonable request. The coordinate and structure factor files are deposited at the PDB (PDB code ID: 7VPY). Raw data are available at Integrated Resource for Reproducibility in Macromolecular Crystallography (https://proteindiffraction.org/). Original images of this article are available in Supplementary Fig. 11. Source data behind the graphs in this article can be found in Supplementary Data 1. Other data are available from the corresponding authors on reasonable request.

## Materials availability

COGNANO Inc. will agree to the use of any materials and methods except for therapeutic use as long as appropriately referring to this paper or the company name; materials will be delivered through NITTOBO MEDICAL Corporation Ltd., Tokyo, Japan; and lateral flow assay (antigen test) kits will be developed and delivered through Yamato Scientific, Co., Ltd., Tokyo, Japan. Responsible requests for materials for research use should be directed to A.T.-K.

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

## Acknowledgements

This study was supported by donations from POPURI Pharmacy Co., Ltd. (Kyoto, Japan), grants from the AMED Research Program on Emerging and Re-emerging Infectious Diseases (JP20fk0108268, JP20fk018517, and JP20fk018413 to A.T.-K.), JSPS KAKENHI (JP20K22630 to J.F. and JP25000013 to Ke.N.), Platform Project for Supporting Drug Discovery and Life Science Research (BINDS) from AMED (JP21am0101117 to Ke.N.), Cyclic Innovation for Clinical Empowerment (CiCLE) from AMED (JP17pc0101020 to Ke.N.), Program on Open Innovation Platform with Enterprises, Research Institute and Academia, Japan Science and Technology Agency (JST, OPERA, JPMJOP1861 to T.I.), JEOL YOKOGUSHI Research Alliance Laboratory of Osaka University to Ke.N and KYOTO industrial Support Organization 21, the subsidies (Sangakukou no Mori) to COGNANO Inc. We thank all staff of Kyoto University Hospital, Kyoto University Medical Research Support Centre of Graduate School of Medicine, especially Yohta Fukuda, Haruyasu Asahara, and Maiko Moriguchi, Osaka University; the BL44XU beamline of SPring-8; and especially Kaori Yurugi and Yoshihito Tsuji, COGNANO Inc. We thank Mikiko Tanaka for the technical assistance. We also thank Dr. Jun-ichi Sakuragi in Kanagawa Prefectural Institute of Public Health for providing SARS-CoV-2. We are pleased to thank Dr. Nakanishi and the veterinarians of KYODOKEN Institute for caring for animals and performing experiments. We also thank the Kyoto University Livestock Farm, which partially maintained the alpacas, and Satoshi Endo, the mayor of Hirono-machi town, Fukushima, and their local government, who granted the construction of a farm to maintain alpacas.

## Author contributions

A.I. and A.T.-K. conceived the study. R.M. performed biochemical experiments, generated large libraries, performed biopanning, and assembled microscopy data. H.Y. performed bioinformatic analyses and collected nasal swab specimens. Yo.K., Ya.K, H.Y., I.A., T.W. and Ka.N. performed virus assays. I.A., K.Y., K.K., Ka.N., Y.Y., K.M., A.R., Ko.S and Ke.S. and Y.M. prepared and certified materials. J.F. prepared cryo-EM specimens and collected data. J.F. and F.M. processed the cryo-EM images. J.F., F.M., T.I. and Ke.N. interpreted structures. K.Y. performed crystallographic studies. R.M., J.F., H.Y., K.K., Ka.N., I.A., A.I., Ke.N. and A.T.-K analysed the data. R.M., J.F., Yo.K., Ya.K., H.Y., I.A., K.Y., K.K. and Ka.N. drew figures. R.M., J.F., H.Y., I.A., K.Y. and A.T.-K. wrote the first draft of the manuscript. J.F., Ko.S., Y.M., T.I., A.I., Ke.N. and A.T.-K. reviewed and commented on the manuscript. All authors approved the reviewed manuscript.

## Competing interests

Kyoto University, Osaka University, and COGNANO Inc. have filed a patent application (JP2021-170471) in connection with this research, on which R.M., J.F., A.T.-K., Ko.S., K.K., H.Y., A.I., F.M., Ke.N., K.Y., T.I., I.A., Y.M., Haruyasu Asahara and Maiko Moriguchi are inventors. A.I. is a stockholder of COGNANO Inc., which has patents and ownership of antibody sequences (JP2021-089414) and an in-house method of identifying antibodies (PCT/JP2019/021353) described in this study on which A.I. is an inventor. R.M., H.Y., and K.K. are employees of COGNANO Inc. The other authors declare no competing interests.
