## [Peer Review File · Communications Biology]

Reviewers' comments:

Reviewer #1 (Remarks to the Author):

The manuscript presents 9 nanobodies that target the receptor binding domain of SARS-CoV-2 spike. The nanobodies were discovered by immunization of alpaca and selection of the enriched clones by panning. The nanobodies were tested also against the alpha, beta, gamma, and delta variants. Overall, multiple RBD nanobodies were previously detected and characterized, including ones with higher affinity and neutralization potency. The main novelty of the current work is the nanobodies (and their epitopes) that can bind RBD and NTD simultaneously. The epitopes were accurately determined combining crystallography and cryo-EM

Can you estimate based on the structures, if the nanobodies will be affected by mutations observed in the Omicron variant?

Related work: there are over 100 published antibodies and dozens of nanobodies. While it is impossible to cite all of them, please cite major works and reviews.

Abstract: lines 42-44: "located inside the NTD" , better say "contacting the NTD"

There are typos in the manuscript (partial list):

Line 54: should be "armour"

Line 64: should be "developed"

Reviewer #2 (Remarks to the Author):

In this study, authors immunized two alpacas with SARS-CoV-2 spike proteins and identified 9 nanobodies that bind to different regions on spike. These nanobodies were then tested for spike binding (FACS, western blot, ELISA, imaging), pseudovirus neutralization, cryo-EM and crystal structure in complex with spike. Interestingly, when binding to down position RBD, clone P86 also interacts with neighboring NTD, and interferes with the ACE2 to access the neighboring up-RBD. Although this binding pattern is novel, it is affected by L452R mutation in SARS-CoV-2 delta variant. Overall, authors made some interesting findings, but also made some claims that are not well supported by their data.

1. Are the selected nine nanobodies the only ones with enrichment scores in the SARS-CoV-2 spike panned sublibraries higher than those in other sublibraries? Are there other criteria involved in candidate selection, such as CDR clustering?
2. Can authors show results of their nanobody-based antigen test on real nasal swab specimens? The detection limit of ELISA seems to be too high for real diagnostic use.
3. Authors claimed that clone P86 targets conserved epitopes on RBD, however, this clone fails to neutralize Delta variant, which is likely due to the L452R mutation near the center of P86 binding interface. L452R mutation is also seen in Omicron variant, which probably will also escape P86's neutralization activity. And there is no western blot data for P86 and several other clones. Therefore, authors need to modify the statement in the abstract that "the clone... maintains activities against spike proteins carrying escape mutations".
4. Authors should test their nanobodies on Omicron variant, for both binding and neutralization. C246 is the only nanobody that can neutralize alpha, beta, gamma and delta variants. It would be

interesting to see whether it continues to neutralize Omicron variant. The combination of E484K and L452R mutations in Omicron variant probably will abolish the neutralization by P86 and P17, disqualifying them for SARS-CoV-2 antibody-therapy.

5. Authors observed 2-RBD-up (2-up+P86) and 3-RBD-up (3-up+P86) from the spike-P86 complex. What is the ratio of these two conformations? Does P86 target the same epitope on RBD at up and down position?

6. Clone C246 could not stain cells overexpressing SARS-CoV-2 spike but can somehow neutralize SARS-CoV-2 pseudovirus. Authors need to test this clone on authentic virus to see whether the neutralization activity is real.

7. Typically, neutralization assay is done by incubating antibody/nanobody with viruses first, followed by infecting cells with antibody/nanobody treated viruses. However, as described in line 744, authors incubated cells with nanobodies first, then infected cells with viruses, which does not seem to make much sense. Can authors clarify this?

Reviewer #1 (Remarks to the Author):

The manuscript presents 9 nanobodies that target the receptor binding domain of SARS-CoV-2 spike. The nanobodies were discovered by immunization of alpaca and selection of the enriched clones by panning. The nanobodies were tested also against the alpha, beta, gamma, and delta variants. Overall, multiple RBD nanobodies were previously detected and characterized, including ones with higher affinity and neutralization potency. The main novelty of the current work is the nanobodies (and their epitopes) that can bind RBD and NTD simultaneously. The epitopes were accurately determined combining crystallography and cryo-EM

We would like to thank the reviewer for finding the novelty of the nanobodies. We are also happy with the reviewer's comment on the accurate determination of the epitopes solved with X-ray crystallography and cryo-EM.

Can you estimate based on the structures, if the nanobodies will be affected by mutations observed in the Omicron variant?

In short, yes. We estimated whether the P86 clone would be affected by which mutations based on the structure of the P86 and SARS-CoV-2 spike Wuhan-1 complex. We concluded that the L452R mutation in the Delta variants would severely affect the binding of the P86; however, any other mutations found in the Omicron variants contribute less to the binding. We have drawn figures (Figure 7) and made tables (Table 1) estimating that the P86 clone would not be affected by any mutated residues of the SARS-CoV-2 Omicron spike, and we discuss this further in the revised manuscript.

Related work: there are over 100 published antibodies and dozens of nanobodies.

While it is impossible to cite all of them, please cite major works and reviews.

Thank you for this comment on related works. In the revised manuscript, we have cited major works and reviews, especially on the SARS-CoV-2 Omicron variants and antibodies as references 31, 45, 56, 52-60, 62-66, and 87.

Abstract: lines 42-44: “located inside the NTD”, better say “contacting the NTD”

There are typos in the manuscript (partial list):

Line 54: should be “armour”

Line 64: should be “developed”

We have corrected the typos, and native English speakers have reviewed the revised manuscript.

Reviewer #2 (Remarks to the Author):

In this study, authors immunized two alpacas with SARS-CoV-2 spike proteins and identified 9 nanobodies that bind to different regions on spike. These nanobodies were then tested for spike binding (FACS, western blot, ELISA, imaging), pseudovirus neutralization, cryo-EM and crystal structure in complex with spike. Interestingly, when binding to down position RBD, clone P86 also interacts with neighboring NTD, and interferes with the ACE2 to access the neighboring up-RBD. Although this binding pattern is novel, it is affected by L452R mutation in SARS-CoV-2 delta variant. Overall, authors made some interesting findings, but also made some claims that are not well supported by their data.

We are thankful to the reviewer for finding the novelty in the binding pattern of the nanobodies. We have addressed these claims by performing experiments. These results are described in the revised manuscript and are written in a point-by-point manner as follows to support our conclusions.

1. Are the selected nine nanobodies the only ones with enrichment scores in the SARS-CoV-2 spike panned sublibraries higher than those in other sublibraries? Are there other criteria involved in candidate selection, such as CDR clustering?

We clustered clones in the panned sublibraries that were not restricted to only CDRs. We have added details in the revised methods section regarding the criteria and results.

Line 540: We clustered nanobody sequences with no more than six Damerau–Levenshtein distances^{84,85}. We found eight and ten clusters—from Cristy’s and Puta’s libraries, respectively—significantly enriched in only the SARS-CoV-2 panned sublibraries. We synthesized the top read clones in each family as linked dimers. Among them, four (C17, C49, C116, and C246) out of eight and five (P17, P86, P158, P334, and P543) out of ten dimer

clones were expressed and further analysed.

2. Can authors show results of their nanobody-based antigen test on real nasal swab specimens? The detection limit of ELISA seems to be too high for real diagnostic use.

We used nasal swab specimens in assays with our nanobody-based antigen test kits. These results have been added to the revised manuscript. We have upgraded our nanobody-based ELISA kit, and assayed using nasal swab specimens via ELISA and western blot.

Line 161: **An ELISA kit, using P158 as the capture nanobody and P543 as the detection nanobody detected the SARS-CoV-2 spike (Fig. 2b).** Both P543 and P86 were able to specifically detect the SARS-CoV-2 spike original and beta variant in cellular homogenates: HEK cells expressing the full-length spikes were lysed and sequentially diluted (Fig. 2c). When the detection antibody was changed to Fc-tagged P543 (P543-Fc), the detection limit was reached below 20 ng ml⁻¹ (Fig. 2d), **and the sensitivity was the same as that of an ELISA kit used for the detection of haemagglutinin of influenza virus^{39,40}.**

Line 178: **Detecting the SARS-CoV-2 Omicron spike variant**

After submission of the original manuscript (10 November 2021), a new SARS-CoV-2 variant (Omicron) was reported (22 November 2021) and declared the fifth VOC (26 November 2021)^{31,45}. The Omicron variant (B.1.1.529 and BA lineages) had become the most recognized VOC and outcompeted the delta variant⁴⁶. Under these circumstances, we assessed whether our nanobodies could detect the SARS-CoV-2 Omicron variant spike protein. First, we tested which nanobodies could detect the Omicron spike protein via western blotting, and we found that P86, C246, P158, P543, and P334 could detect both the original and Omicron variant spike proteins (Fig. 3a). Second, using flow cytometry assays, we reconfirmed that P17, P86, P158, P334, and P543 could recognize the Omicron spike on the cell surface (Fig. 3b). Therefore, we tested the ability of the kits made with nanobodies to detect SARS-CoV-2 spike proteins from nasal swab specimens.

Nanobody-based test kits for real nasal swab specimens

We sampled 22 swab specimens from emergency fever patients. Sixteen samples were confirmed to be SARS-CoV-2 positive by PCR. We compared these samples to samples from healthy volunteers with P158-based western blotting and observed smeared and accumulated bands in swab specimens from the patients (Fig. 4a). The P158 nanobody detected bands in the 16 patient samples (Fig. 4b) with molecular sizes similar to those of transfected spike proteins (Fig. 3a).

Next, we developed nanobody-based lateral flow assay kits for nasal swab specimens. We

used P158-dimer for the capture line and conjugated the P543-trimer to fluorescent beads. We sampled swab specimens by adding detergents and nuclease (see Methods). We observed positive lines in the nasal swab specimens from the patients but not in those from the healthy volunteers (Fig. 4c). For the detection of spike proteins in nasal swab specimens using sandwich ELISA, we used a Flag-tagged P86-trimer as the detection antibody (the capture nanobody P158-dimer was His-tagged). The detection limit was below 16 ng ml^{-1} . Thus, this nanobody-based ELISA kit was able to detect spike proteins in the nasal swab specimens from patients (Fig. 4d).

3. Authors claimed that clone P86 targets conserved epitopes on RBD, however, this clone fails to neutralize Delta variant, which is likely due to the L452R mutation near the center of P86 binding interface. L452R mutation is also seen in Omicron variant, which probably will also escape P86's neutralization activity. And there is no western blot data for P86 and several other clones. Therefore, authors need to modify the statement in the abstract that "the clone... maintains activities against spike proteins carrying escape mutations".

We would like to thank the reviewer for their concern regarding the SARS-CoV-2 Omicron variant. We confirmed that the Omicron variants (BA.1, BA.2, and BA.3) have more than 30 mutations. However, almost all of the variants do not have L452R mutation. We

examined the GISAID database and found that less than 0.6% of the deposited Omicron sequences have the L452R mutation (2022-2-2).

Regarding the western blot data, we tested the clones for the Omicron variant via western blot, and these results are shown in the revised manuscript (Figure 3). In short, P86, C246, P543, and P158 detected the SARS-CoV-2 Omicron variant spike, whereas C17 and C116 did not.

Therefore, we rephrased the abstract and deleted the sentence that *“the clone… maintains activities against spike proteins carrying escape mutations”*, because escape mutations have become varieties after the emergence of the Omicron variants.

Line 37: Abstract

We are amid the historic coronavirus infectious disease 2019 (COVID-19) pandemic. Imbalances in the accessibility of vaccines, medicines, and diagnostics among countries, regions, and populations, and those in war crises, have been problematic. Nanobodies are small, stable, customizable, and inexpensive to produce. Herein, we present a panel of nanobodies that can detect the spike proteins of five SARS-CoV-2 variants of concern (VOCs) including Omicron. We show via ELISA, lateral flow, kinetic, flow cytometric, microscopy, and western blotting assays that our nanobodies can quantify the spike variants. This panel

of nanobodies broadly neutralized viral infection caused by pseudotyped **and authentic** SARS-CoV-2 VOCs. Structural analyses showed that the P86 clone targeted epitopes that were conserved yet unclassified on the receptor-binding domain (RBD) and **contacted** the N-terminal domain (NTD). Human antibodies **rarely** access both regions; consequently, the clone buries hidden crevasses of SARS-CoV-2 spike proteins **that go** undetected by conventional antibodies.

4. Authors should test their nanobodies on Omicron variant, for both binding and neutralization. C246 is the only nanobody that can neutralize alpha, beta, gamma and delta variants. It would be interesting to see whether it continues to neutralize Omicron variant. The combination of E484K and L452R mutations in Omicron variant probably will abolish the neutralization by P86 and P17, disqualifying them for SARS-CoV-2 antibody-therapy.

We assessed the binding of the clones to the Omicron variant via western blotting and FACS (Figure 3). We examined their neutralizing activity using pseudotyped viruses developed by different investigators. Consequently, we found that P86 bound and neutralized the Omicron variant with the same efficiency as the original SARS-CoV-2 spike. We have added these results to the revised Figure 8 and Figure 9.

Furthermore, we added figures describing the mutations found in the Omicron variant (Figure 7) and referred to the works and reviews on the variants.

Line 267: **Epitope mapping of P86 to the Omicron variant**

The Omicron variants have spread rapidly. To date, several Omicron lineages have been classified: BA.1, BA.2, and BA.3^{46,52} (Fig. 7). Recent works warn that almost all neutralizing antibodies on the market or under development do not have the activity against the Omicron variants, except for sotrovimab (S309), which maintains the neutralizing activity against BA.1⁵³⁻⁵⁷. Additionally, several research groups have reported that the neutralizing activity of sotrovimab is significantly reduced against BA.2^{58,59}.

We mapped the mutations of the Omicron variants on the 2-up+P86 structure (Fig. 7a,b) and evaluated how each mutation would affect the nanobody binding^{60,61} (Table 1). While the L452R mutation in the Delta variant was the most effective in reducing the nanobody binding ($\Delta G = +1.0$ kcal mol⁻¹), any mutations to BA.1 and BA.2 (including BA.3) would not affect the binding of P86. In particular, the variants with Q493R, G496S, and Q498R mutations bound to ACE2^{62,63} were covered by P86 without contributing to the binding (Fig. 7c and Table 1). Finally, P86 also contacted the inner side of the NTD, where no mutations have been seen in any VOCs (Fig. 7a,b).

Neutralizing activity of the nanobodies against the SARS-CoV-2 Omicron variants

We assessed whether P86 neutralizes the SARS-CoV-2 Omicron variants using pseudotyped and authentic viruses. As a control, we included sotrovimab (S309) in these assays. P86 neutralized both the BA.1 and BA.2 Omicron pseudotyped viruses, while sotrovimab showed a reduction in its ability to suppress the BA.2 Omicron variant (Fig. 8a,b).

Line 77: Moreover, the epitopes of P86 are conserved among the Omicron variants BA.1, BA.2, and BA.3. P86 potently neutralized the SARS-CoV-2 Omicron variants compared to clinically available therapeutic antibodies.

Line 323: After submitting the original manuscript, the Omicron variants have emerged, spread, and outcompeted the Delta variant. The Omicron variants escaped from the sera of vaccinated or previously infected people and evaded almost all therapeutic antibodies. However, we presented a panel of nanobodies that detected the Omicron spike variants in nasal swab specimens and neutralized the SARS-CoV-2 Omicron variants.

5. Authors observed 2-RBD-up (2-up+P86) and 3-RBD-up (3-up+P86) from the

spike-P86 complex. What is the ratio of these two conformations? Does P86 target the same epitope on RBD at up and down position?

We have added the ratios of 2-up+P86 (46.2%) and 3-up+P86 (53.8%) to the text.

We reanalyzed the epitopes on RBD at the up and down positions using Protein Interfaces, Surfaces, and Assemblies (PISA: E. Krissinel and K. Henrik (2007). Inference of macromolecular assemblies from crystalline state J. Mol. Biol. 372, 774-797). The results were so informative and well correlated to the experimental data that we added tables and rewrote the results section.

Line 275: We mapped the mutations of the Omicron variants on the 2-up+P86 structure (Fig. 7a,b) and evaluated how each mutation would affect the binding^{60,61} (Table 1). While the L452R mutation in the Delta variant was the most effective ($\Delta G = +1.0 \text{ kcal mol}^{-1}$), any mutations to BA.1 and BA.2 (including BA.3) would not affect the binding of P86. In particular, the variants with Q493R, G496S, and Q498R mutations bound to ACE2^{62,63} were covered by P86 without contributing to the binding (Fig. 7c and Table 1). Finally, P86 contacted the inner side of the NTD, where no mutations have been seen in any VOCs (Fig. 7a,b).

6. Clone C246 could not stain cells overexpressing SARS-CoV-2 spike but can somehow neutralize SARS-CoV-2 pseudovirus. Authors need to test this clone on authentic virus to see whether the neutralization activity is real.

We tested the C246 and P86 clones on an authentic virus at an independent laboratory. We confirmed that C246 had reduced neutralization activity and P86 retained its activity. We also compared these activities to that of sotrovimab (S309) from GlaxoSmithKline. These results have been added to Figure 8 and Figure 9. Additionally, in the discussion section, we rewrote why C246 showed reduced activity against an authentic virus. In short, there are two ways to invade cells: fusion and endocytosis. Although a pseudovirus infects cells only once, authentic viruses amplify, transmit, and reinfect cells.

Line 290: Finally, we evaluated the neutralizing activities of P86 and C246 using the authentic Omicron BA.1 variant virus (strain hCoV-19/Japan/TY38-873/2021) (Fig. 9). Although C246 could not neutralize the virus even at the highest concentration tested ($25 \mu\text{g ml}^{-1}$) (Fig. 9c), P86 suppressed plaque formation of the authentic Omicron BA.1 variant virus (IC_{50} of $0.18 \mu\text{g ml}^{-1}$) with an efficiency similar to that of sotrovimab (IC_{50} of $0.52 \mu\text{g ml}^{-1}$) (Fig. 9a,b).

Line 304: A limitation of this study is that the C246 clone could not neutralize the authentic SARS-CoV-2 virus infection. These results signify that antibodies and chemicals neutralizing pseudotyped viruses cannot necessarily neutralize real viruses⁶⁴⁻⁶⁶.

7. Typically, neutralization assay is done by incubating antibody/nanobody with viruses first, followed by infecting cells with antibody/nanobody treated viruses. However, as described in line 744, authors incubated cells with nanobodies first, then infected cells with viruses, which does not seem to make much sense. Can authors clarify this?

We would like to thank the reviewer for pointing this out. We have corrected the method as follows.

Line 762: Pseudotyped virus neutralization assay

Fivefold sequentially diluted nanobodies were incubated with SARS-CoV-2 pseudotyped viruses for one hour. K562 and HOS cells expressing human ACE2 and TMPRSS2 were subsequently infected with the antibody-virus mixture for one hour at 37 °C and cultured for two days.

Fig. 3| Nanobodies against the SARS-CoV-2 Omicron variant.

a, Nanobody-based western blotting assays of SARS-CoV-2 spikes. HEK cell lysates (C: as control of mock transfection) expressing C-terminally C9-tagged SARS-CoV-2 spike proteins (W: Wuhan-1) and Omicron variant (O: Omicron BA.1) were blotted with an anti-C9 antibody and the indicated nanobodies. **b**, Flow cytometric analyses of DT40 cells. Log scaled signals (x-axis) of the indicated eight nanobodies with parent cells (black) and cells stably expressing the SARS-CoV-2 Omicron spike protein (red) were counted (y-axis) and plotted.

Figure 3

Fig. 4| Nanobodies for immune assays testing of nasal swab specimens.

a, P158 nanobody-based western blotting of nose swab specimens. Patients with fever (+) and confirmed to be positive for SARS-CoV-2 by PCR test (+); healthy volunteers with no fever (-) and negative by PCR test (-) were compared (NT: not tested). **b**, P158 nanobody-based western blotting assays of swabs from emergency fever patients. A to D and I to P were confirmed positive for SARS-CoV-2 by PCR test; E to H were not confirmed to be positive by the PCR test. **c**, P158 nanobody-lined lateral flow assays of swabs from emergency fever patients. Control samples and the patient ID are the same as those in (a) and (b), respectively. **d**, P158 nanobody-captured ELISA. Detection signals of the nasal swab specimens are shown (n=2).

Figure 4

BA.1: A67V, H69-V70del, T95I, G142D, V143-Y145del, N211I, L212del, R214EPE, G339D, S371L, S373P, S375F, K417N, N440K, G446S, S477N, T478K, E484A, Q493R, G496S, Q498R, N501Y, T547K

BA.2: T19I, L24S, P25-P26del, A27S, G142D, V213G, G339D, S371F, S373P, S375F, T376A, D405N, R408S, K417N, N440K, S477N, T478K, E484A, Q493R, Q498R, N501Y, Y505H

Fig. 7| Fitting model of P86 with the SARS-CoV-2 Omicron variants.

a,b, The down-RBD, the neighbouring NTD, and P86 are coloured grey, cyan, and pink, respectively. Spike protein mutations of the SARS-CoV-2 Omicron variants BA.1 (**a**) and BA.2 (**b**) are depicted in red. **c,** A close-up view of the binding domain of P86 with the RBD denoting residues E484, Q493, G496, and Q498.

Figure 7

Fig. 7 | Fitting model of P86 with the SARS-CoV-2 Omicron variants.

a,b, The down-RBD, the neighbouring NTD, and P86 are coloured grey, cyan, and pink, respectively. Spike protein mutations of the SARS-CoV-2 Omicron variants BA.1 (**a**) and BA.2 (**b**) are depicted in red. **c,** A close-up view of the binding domain of P86 with the RBD denoting residues E484, Q493, G496, and Q498.

Figure 7

Figure 8

Fig. 8| Pseudotyped virus neutralization assays.

a, Means of the number of infectious units of virus for the SARS-CoV-2 spike Omicron variants (n=3). HOS cells expressing ACE2 and TMPRSS2 infected with HIV-based pseudotyped virus were assessed. Relative infection units at the indicated concentrations of the P86 and sotrovimab are plotted. **b**, Summary of the trends in the neutralizing activities of P86 and sotrovimab and the measured IC₅₀s for the SARS-CoV-2 Omicron variants BA.1 and BA.2.

Fig. 9| Authentic virus neutralization assays.

a,b, Concentration dependency of the P86 nanobody (**a**) and sotrovimab antibody (**b**) for plaque reduction of the SARS-CoV-2 Omicron BA.1 variant ($n=2$). The ratios of plaque formation are expressed as a percentage of the plaque numbers from untreated cells (means \pm SD) and fitted with a sigmoidal function using Igor Pro (ver. 8.04, Wavemetrics). **c,** The C246 nanobody versus plaque formation of the Wuhan and Omicron BA.1 variants ($n=1$). The ratios of plaque formation are shown as a percentage of the plaque numbers from untreated cells.

Figure 9

Table 1 | Comparison of the epitopes of Up-RBD+P86 and Down-RBD+P86 via Protein Interface, Surfaces, and Assemblies (PISA).

No. residues		Accessible Surface Area (Up)	Buried Surface Area and Bonds (H, S)* (Up)	Solvation Energy Effect: ΔG (Up)	Accessible Surface Area (Down)	Buried Surface Area and Bonds (H, S)* (Down)	Solvation Energy Effect: ΔG (Down)
	BA.1+BA.2	\AA^2	\AA^2	kcal mol ⁻¹	\AA^2	\AA^2	kcal mol ⁻¹
344	Ala	25.1	0	0	31.53	0	0
345	Thr	110.91	0	0	124.78	0	0
346	Arg	173.84	77.26 (S)	-1.54	192.26	72.1	-0.75
347	Phe	21.45	9.37	-0.08	10.98	5.73	-0.02
348	Ala	28.97	26.55	0.37	15.75	15.75	0.25
349	Ser	11.77	9.43 (H)	-0.08	3.61	3.61	0.25
350	Val	0	0	0	0	0	0
351	Tyr	74.5	62.49	-0.03	78.1	70.74 (H)	0.27
352	Ala	48.52	48.02 (H)	0.57	9.91	5.39	0.03
353	Trp	30.81	26.59	0.08	9.86	0.49	-0.01
354	Asn	64	32.34 (H)	-0.23	60.18	27.1	-0.05
355	Arg	103.72	18.72	0.14	71.68	5.51	-0.06
356	Lys	51.9	0	0	91.92	0	0
357	Arg	144.31	0	0	141.1	0	0
445	Val	154.15	0	0	137.36	0	0
446	Gly	20.66	0	0	11.42	0	0
447	Gly	26.79	8.7	0.14	32.48	0	0
448	Asn	3.92	3.92	-0.04	95.95	36.09 (H)	0.13
449	Tyr	86.46	29.68	0.38	45.61	7.23	0
450	Asn	20.7	13.06	-0.15	33.66	25.14	-0.26
451	Tyr	116.94	76.62	0.96	47.31	39.99 (H)	0.56
452	Leu	43.32	37.5	0.6	64.98	62.3 (H)	1
453	Tyr	39.66	0	0	36.07	0	0

454	Arg	21.47	0	0	4.62	0	0
464	Phe	80.55	0	0	50.92	0	0
465	Glu	59.93	0	0	42.78	0	0
466	Arg	3.58	0.44	-0.01	155.78	86.19 (H, S)	-1.91
467	Asp	123.2	23.66	0.02	38.29	0	0
468	Ile	79.87	47.65	0.38	69.49	43.98	0.7
469	Ser	58.67	1.34	0.02	57.58	0	0
470	Thr	47.03	37.4	0.09	49.99	39.84	0.36
471	Glu	117.88	65.91 (H)	0.15	140.23	104.47 (H)	-0.62
472	Ile	32.26	1	0.02	34.84	0	0
473	Tyr	51.04	0	0	20.84	0	0
474	Gln	71.55	0	0	13.1	0	0
475	Ala	32.8	0	0	30.71	0	0
476	Gly	4.57	0	0	62.89	0	0
477	Ser	95.29	0	0	86.67	0	0
478	Thr	123.99	0	0	34.24	0	0
479	Pro	65.07	0	0	82.4	0	0
480	Cys	101.19	0	0	95.47	6.5 (H)	-0.07
481	Asn	122.21	46.78 (H)	-0.54	95.46	1.67	0.03
482	Gly	47.69	35.42 (H)	-0.07	80.11	59.16	-0.3
483	Val	117.5	103.34 (H)	1.39	45.42	31.64 (H)	0.08
484	Glu	11.41	5.08	-0.05	101.28	29.07 (H)	0.02
485	Gly	70.05	37.63 (H)	0.16	0	0	0
486	Phe	116.09	0	0	0	0	0
487	Asn	84.63	0	0	0	0	0
488	Cys	56.1	29.74	0.13	38.53	5.76	0.24
489	Tyr	179.1	3.17	0.05	0	0	0
490	Phe	11.21	6.25	0.05	92.52	72.76	1.16
491	Pro	13.66	0	0	0	0	0
492	Leu	88.49	79.6	1.13	53.01	43.04	0.41
493	Gln	98.13	75.32 (H)	0.31	60.1	19.21	-0.08
494	Ser	96.92	6.52	0.1	74.39	37.03	0.14
495	Tyr	13.89	13.89	-0.15	24.32	11.03	-0.13
496	Gly	30.17	12.9	0.21	28.5	3.84	0.06
497	Phe	14.23	13.45 (H)	-0.15	12.94	12.94 (H)	0

498	Gln	93.16	37.94	0.06	75.68	24.12	-0.24
499	Pro	71.07	2.68	0.04	0	0	0
500	Thr	104.74	0	0	0	0	0
501	Asn	18.17	0	0	0	0	0
502	Gly	47.26	0	0	0	0	0
108	Thr				0	0	0
109	Thr				142.27	8.75	0.06
129	Lys				0	0	0
130	Val				56.73	13.39	0.21
158	Arg				175.29	0	0
159	Val				18.68	0	0
160	Tyr				94.9	22.88	0.37
161	Ser				97.2	4.17	-0.02
162	Ser				165.42	106.89 (H)	0.2
230	Pro				94.51	0	0
231	Ile				18.32	2.87	-0.03
232	Gly				63.8	11.33 (H)	-0.13
233	Ile				54.69	9.66	0.15
234	Asn				104.25	0	0

Bonds*: hydrogen bond (H) and salt bridge (S)

The R346K mutation is seen in the BA.1⁺ lineages. Residues 108 to 234 are in the NTD.

REVIEWERS' COMMENTS:

Reviewer #1 (Remarks to the Author):

The authors addressed all the comments

Reviewer #2 (Remarks to the Author):

It is nice to see that authors have made a lot of effort to make their work more relevant to the real world, by providing data on Omicron variant and nasal swab samples. Authors have well addressed my other concerns too.

There is one minor point authors may want to correct in the reporting summary. Information about nasal swab needs to be included in "Sample size" section.

Reviewer #1 (Remarks to the Author):

The authors addressed all the comments

We thank the Reviewer.

Reviewer #2 (Remarks to the Author):

It is nice to see that authors have made a lot of effort to make their work more relevant to the real world, by providing data on Omicron variant and nasal swab samples. Authors have well addressed my other concerns too.

There is one minor point authors may want to correct in the reporting summary. Information about nasal swab needs to be included in "Sample size" section.

We included information about the nasal swab specimen in the Sample size section of the reporting summary.

We are grateful to the Reviewers for improving the manuscript.